# Ecological landscapes guide the assembly of optimal microbial communities

**Ashish B. George** [1,2]*, **Kirill S. Korolev** [1,3]*

**1** Department of Physics and Biological Design Center, Boston University, Boston, Massachusetts, United States of America, **2** Carl R. Woese Institute for Genomic Biology and Department of Plant Biology, University of Illinois at Urbana-Champaign, Urbana, Illinois, United States of America, **3** Graduate Program in Bioinformatics, Boston University, Boston, Massachusetts, United States of America

* ashish.b.george@gmail.com (ABG); korolev@bu.edu (KSK)

**Data Availability Statement:** The code used to generate and analyze the data is available at https://github.com/ashish-b-george/ecological-landscapes.git.

## Abstract

Assembling optimal microbial communities is key for various applications in biofuel production, agriculture, and human health. Finding the optimal community is challenging because the number of possible communities grows exponentially with the number of species, and so an exhaustive search cannot be performed even for a dozen species. A heuristic search that improves community function by adding or removing one species at a time is more practical, but it is unknown whether this strategy can discover an optimal or nearly optimal community. Using consumer-resource models with and without cross-feeding, we investigate how the efficacy of search depends on the distribution of resources, niche overlap, cross-feeding, and other aspects of community ecology. We show that search efficacy is determined by the ruggedness of the appropriately-defined ecological landscape. We identify specific ruggedness measures that are both predictive of search performance and robust to noise and low sampling density. The feasibility of our approach is demonstrated using experimental data from a soil microbial community. Overall, our results establish the conditions necessary for the success of the heuristic search and provide concrete design principles for building high-performing microbial consortia.

## Author summary

Research shows that microbial communities comprised of specific species combinations can cure disease, improve agricultural output, or synthesize valuable chemicals. But finding the species combinations that generate high-performing communities is challenging because there are too many species combinations to test exhaustively. So, scientists use heuristic strategies that test only a few species combinations to search for high-performing communities. However, these heuristic strategies often fail to find the best species combinations, and we still do not understand when they fail. Here, we develop a framework to analyze these heuristic strategies, building on the concept of fitness landscapes studied in evolution and computer science. We apply this framework to data from simulated

**Funding:** This work was supported by the Simons Foundation Grant #409704, Cottrell Scholar Award #24010 from the Research Corporation for Science Advancement, and by NIGMS grant #1R01GM138530-01 to K.S.K. The funders had no role in study design, data collection and analysis, decision to publish, or preparation of the manuscript.

**Competing interests:** The authors have declared that no competing interests exist.

microbial community models to identify biological properties that affect the success of heuristic search strategies, such as the extent to which microbes compete for the same metabolites. Further, we establish statistical measures of the landscape structure that can help estimate search success from preliminary data. We validate our findings using experimental data from communities of soil microbes. Together, our results develop a conceptual framework to analyze and develop heuristic search strategies and identify guiding principles to help scientists choose species and environmental conditions that make finding high-performing microbial communities easier.

## Introduction

Life on earth is sustained by a myriad of biochemical transformations. From synthesis to decomposition, microbial communities perform the bulk of these transformations, including photosynthesis, nitrogen fixation, and the digestion of complex molecules [1–3]. This understanding has generated considerable interest in exploiting microbial capabilities for producing energy and food, degrading waste, and improving health [4–10].

The metabolic ingenuity of microbial communities can be harnessed by simply placing the right combination of species in the right environment. Yet, the search for the appropriate species and environmental conditions is anything but simple. Ecological interactions are nonlinear and often unpredictable, so an exhaustive search across all relevant variables is the only sure way to build a community with the best performance. Such an exhaustive search is however infeasible because the number of possible combinations increases exponentially with the number of species and environmental factors; just 16 species leads to over 65, 000 combinations. Even with expert knowledge, computer simulations, and liquid-handling robots, exhaustive testing remains infeasible for any complex microbial community [11–19].

This challenge is not unique to ecology. In fact, most optimization problems cannot be solved directly, and one has to resort to heuristic search strategies [20, 21]. For microbial communities, a simple heuristic search (a greedy gradient-ascent) proceeds via a series of steps. At each step, a set of new communities is created by adding or removing one or a few species from the current microbial community. The community with the best performance is then chosen for the next step. Although easy to implement, the search can get stuck at a local optimum and achieve only a small fraction of the best possible performance. It is therefore essential to identify when such heuristic search is likely to succeed.

Here, we study the success of the heuristic search by simulating a range of realistic and widely-used *consumer-resource* models of microbial communities, which have reproduced many features of natural and experimental communities [22–24]. Compared to the commonly-used Lotka-Volterra models, our approach is more realistic and makes a closer connection to the metabolic interactions that underpin many optimization problems in biotechnology [25–29]. Furthermore, it is much easier to make educated guesses about the production and consumption of resources than interaction coefficients. A shortcoming of these models is that they neglect non-resource mediated interactions such as pH or antimicrobials, or sequential uptake of resources. Although such mechanisms play a role in natural communities, it is unclear, at present, how to incorporate them into a general modeling framework. Further, recent work shows that resource competition is the predominant mode of interactions in *in vitro* gut bacterial communities [30].

In most of our simulations, microbes compete for externally supplied resources, but we also test the robustness of our conclusions by allowing microbes to produce other metabolites that

can be then consumed by every member of the community. We found how to vary the structure of interspecific interactions and determined its effects on the efficacy of the search.

To integrate our findings into a coherent framework and understand when search fails, we define and analyze community function landscapes utilizing an analogy to the fitness landscapes from population genetics [31]. The community function of interest is analogous to fitness, and the composition of the community is analogous to the genome. The heuristic search then corresponds to an uphill walk on this multi-dimensional landscape. Similar to previous studies in evolution, we found that search success depends on the *ruggedness* of the ecological landscape. Multiple definitions of ruggedness exist, and we identified those that are more predictive across diverse ecological scenarios and can be reliably estimated from limited, noisy experimental data. The intuition gained from our numerical studies applied well to real experimental data on an ecological landscape for six soil microbes from Ref. [32].

Overall, our work identifies the connections between ecological properties, community structure, and optimization. This knowledge makes it possible to predict whether heuristic search is feasible and provides concrete strategies to increase the chance of success by adjusting the environment or the pool of candidate species. Furthermore, we anticipate that community landscapes can provide a unifying framework to compare microbial consortia and their assembly across diverse ecological settings.

## Model

In consumer-resource models, growth of each microbe depends on its ability to consume various available resources and convert consumed resources into biomass. The ability of species to consume different resources is encoded in the consumption matrix, which distinguishes the different species (Fig 1). Typically, the resources are externally supplied, but, when we consider metabolite leakage, there is a single externally supplied resource and most members of the community rely on the metabolic byproducts leaked by other members of the community [33–36]. Both the species and the resources are diluted at a fixed rate to prevent the accumulation of nutrients and biomass, as in a chemostat. The mathematical details of the models are further explained in Methods. Numerical simulations were carried out using code based on the 'Community Simulator' package [37].

Ecological dynamics, including those of microbial communities, could be very complex and include chaos, oscillations, sensitivity to initial conditions due to multistability etc. In the context of optimizing a community function, one favors a community with much simpler dynamics: typically a stable equilibrium that is robust to perturbations. Moreover, this simple scenario is the one most frequently observed in experiments with a handful of species and in many types of ecological models (at least in certain regions of the parameter space); see Table 1 and Ref. [30, 38–40]. This is indeed the case for consumer-resource models, which always reaches a unique steady-state that depends only on the initial presence or absence of species [41].

The search for the optimal community begins with a choice of the desired ecological function, denoted as $\mathcal{F}$; this could be the production rate of a particular metabolite, the degradation rate of an undesirable compound, or even the diversity of the community. One also needs to choose the pool of $S$ candidate species from which to assemble the microbial community. In total, there are $2^S$ possible communities, corresponding to distinct species combinations that can be labeled by $\vec{\sigma}$, a string of ones and zeros denoting the presence or absence of each species in the starting community (Fig 1). The simplest search protocol starts with one of the $2^S$ possibilities, for e.g., the community with all species present. Then, every step of the search is a

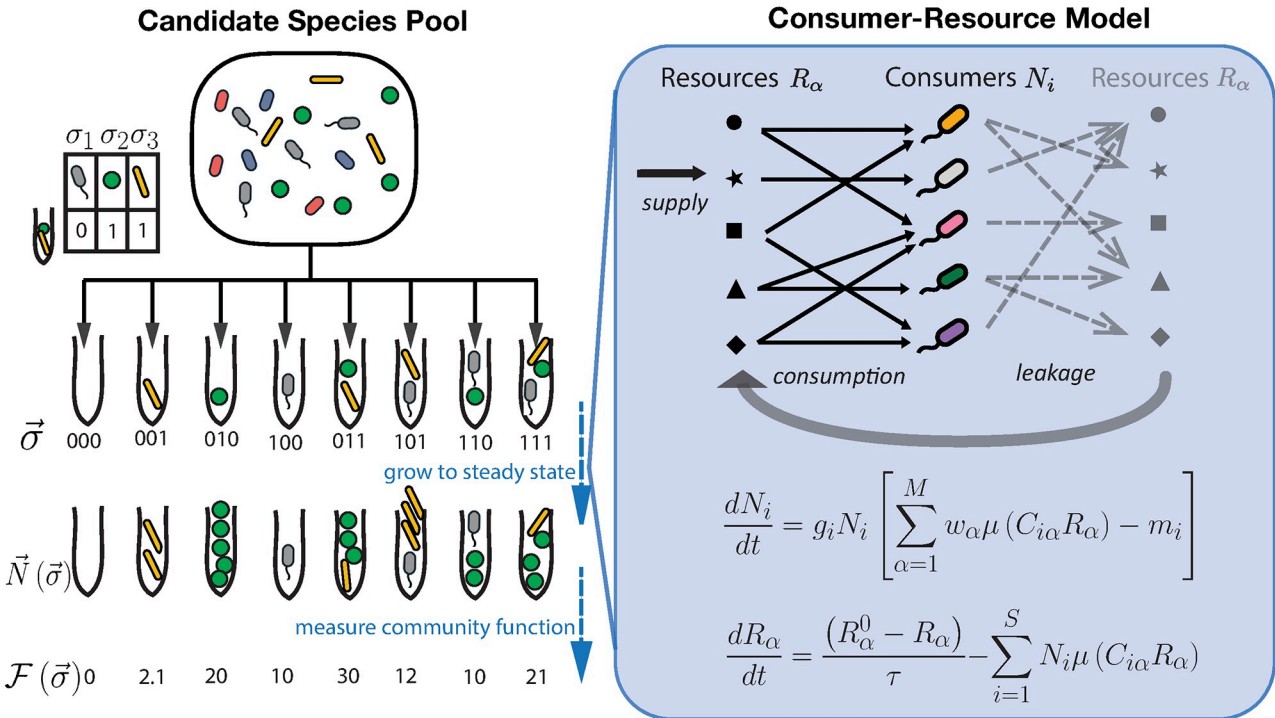

**Fig 1. An ecological landscape framework for community optimization.** We consider the problem of assembling an ecological community from a pool of *S* candidate species that is optimal at performing a desired function. The candidate species in the pool are indexed from 1 to *S*. A test community is seeded with a subset of the candidate species. The composition of the community is encoded in $\vec{\sigma}$ using 0 for absence and 1 for presence for each of the species. The test communities are allowed to grow until they reach a steady state. The species abundances in the steady-state community is $\vec{N}(\vec{\sigma})$, and the community function is $\mathcal{F}(\vec{\sigma})$. The growth of the community to steady-state is simulated by consumer-resource models, illustrated on the right. The growth rate depends on its ability to consume resources and death occurs by dilution or mortality. In models with leakage (gray), metabolic byproducts can be leaked and used by other species. Equations shown correspond to the scenario without metabolic leakage; see methods for further explanation and model equations with leakage.

heuristic gradient-ascent, which considers the current community and the *S* neighboring communities obtained by changing each of the elements of $\vec{\sigma}$, i.e., adding or removing a species. These communities are simulated until they reach a steady state, and the community with the highest value of $\mathcal{F}$ is taken to be the next current state. The process repeats until the community function cannot be improved any further by adding or removing a single species. We also explore several variations of this search protocol after obtaining the main results in the context of the search protocol just described. Mathematically, we can understand the above as the optimization by gradient ascent search of the objective function $\mathcal{F}(\vec{\sigma})$, where $\vec{\sigma}$ takes values in $\{0, 1\}^S$.

Because consumer-resource models reach a unique steady state, all quantities that depend on species and resource concentrations are uniquely determined by the presence-absence of species at the beginning of the simulation. Thus, the community function is fully determined by $\vec{\sigma}$. The search for the optimal community then reduces to the maximization of $\mathcal{F}$ over $\vec{\sigma}$, and $\mathcal{F}(\vec{\sigma})$ can be viewed as a multi-dimensional ecological or community function landscape. The landscape can also be related to the fitness landscapes studied in population genetics [31], with $\mathcal{F}$ analogous to fitness and $\vec{\sigma}$ analogous to genotype. Below, we explore the structure of ecological landscapes under different ecological

**Table 1. Microbial communities often converge to a unique stable steady state, but more complex dynamics are possible.** By a single stable steady-state, we mean that the dynamics converge to a steady-state that is stable to invasion by species that had gone extinct *en route*. In other words, if a species present initially goes extinct and is absent from the final steady-state, then it can not invade the final community successfully. Thus species presence-absences in the initial community fully determine the outcome. Note that this table only approximately condenses a vast and complex literature; for a more thorough understanding we encourage the interested reader to consult the cited references. We have provided additional details regarding the cited studies in S1 Text Sec.1. Macroscopic organisms are discussed in Ref. [64].

| Outcome of dynamics | Models and experiments |
|---|---|
| Single stable steady state for defined starting species | Consumer-resource models with substitutable resources: [41–44]. Consumer-resource models with cross-feeding or thermodynamic constraints: [41, 44, 45]. Many Lotka-Volterra models including those with random interactions with low to moderate variance: [46–49] Compatible empirical observations: [50–54]. |
| Typically a single steady state (multiple steady states may occur in a few percent of initial conditions) | Consumer resource models with non-substitutable resources or species consuming resources diauxically: [55, 56]. Compatible empirical observations: [57]. |
| Complex dynamics and chaos | Stochastic neutral models: [58]. Consumer resource models with highly nonlinear uptake of more than 3 non-substitutable resources: [59, 60]. Lotka-Volterra models with random interactions with large variance or population sub-division: [46, 61]. Compatible empirical observations: [62, 63]. |

conditions and examine the influence of landscape structure on the success of the heuristic search.

## Results

### Niche overlap controls the complexity of ecological landscapes

The problem of finding optimal microbial communities brings a number of questions about the landscape structure and its influence on search success. Before we begin to explore these questions in detail, it is convenient to establish how one can control the difficulty of the search by varying a relevant ecological variable. Intuition suggests that communities with many inter-specific interactions should be more complex than communities where species are largely independent from each other. Therefore, we explored how the density of metabolic interactions influences community structure.

The density of resource-mediated interactions is controlled by the overlap in the resource utilization profiles of the species in the community. Simply put, when more species consume any given resource, the niche overlap and number of competitive interactions increases. In our model, species interactions are encoded in the consumption matrix; see Fig 2A. We created these matrices by allowing each species to consume only a random subset of $M_{consumed}$ resources out of $M_{total}$ resources supplied externally. Since we vary only the types of resources consumed, the niche overlap could be quantified by the number or fraction of resources that both species could consume. More generally, the niche overlap could be computed from the consumption matrix e.g. as the average cosine similarity between the consumption preferences of each species (see S1 Fig), which would take into account the differences in the consumption rates of the various resources.

When $M_{consumed} \approx M_{total}$, the species are generalists; several species are consuming each of the resources, and the niche overlap is high (Fig 2B). In the opposite limit ($M_{consumed} \ll$

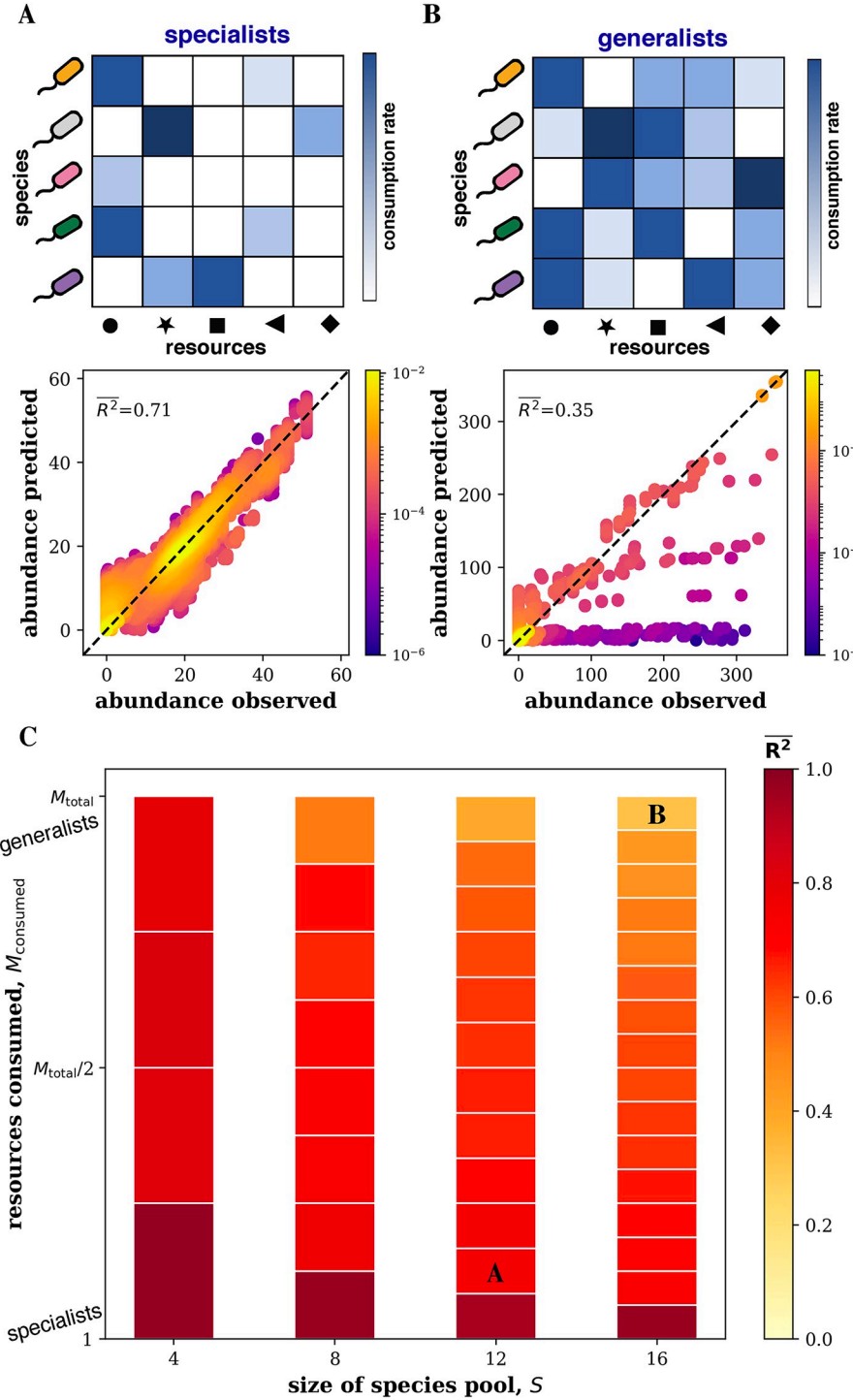

**Fig 2. Niche overlap determines the linearity of ecological landscapes.** The top panels in (A) and (B) show the consumption matrices for species in each species pool. In (A), the niche overlap is low because the species are specialists and consume only a few resources. In (B), the niche overlap is high because the species are generalists and consume many resources. The bottom panels show how well a linear model (Eq 1) fits the steady-state abundances across all possible species combinations. Color bar quantifies the local density of points, as measured by gaussian kernel density estimation. The goodness of the model fit, quantified by $\overline{R^2}$, is higher when the niche overlap is lower. This conclusion is robust to varying the size of the community and the degree of niche overlap. (C) shows how $\overline{R^2}$ varies with the number of species in the pool, $S$, and the average number of resources that they consume, $M_{consumed}$. The degree of niche overlap is determined by $M_{consumed}/M_{total}$. Letters indicate the parameters used in panels A and B.

Note that $\overline{R^2}$ was computed by averaging $R^2$ of the linear model fitted separately for each species, and the number of resources supplied was fixed to $M_{\text{total}} = S$. For each set of parameters, results were averaged over 10 independently generated species pools. Simulations were of consumer-resource models without leakage. Other simulation parameters are provided in Methods, and the robustness of the conclusions to certain modeling assumptions is further discussed in S1 Text and S2 Fig.

$M_{\text{total}}$), species are specialists; most species are consuming different resources, and the niche overlap is low. Thus, $M_{\text{consumed}}/M_{\text{total}}$ controls the density of competitive interactions and could influence the ruggedness of the community landscape.

To gain a broader view of the community structure, we opted not to consider a specific ecological function, but instead looked at the abundances of all species in the community. Thus, we studied the assembly map, $\vec{N}(\vec{\sigma})$, from initial composition to steady-state abundances. Since optimization is easy for linear problems, we quantified the linearity of $\vec{N}(\vec{\sigma})$ by fitting the following model to the data from the consumer-resource models at steady-state

$$\hat{N}_i(\vec{\sigma}) = \begin{cases} \sum_{j=1}^{S} A_{ij}\sigma_j, & \text{if } \sigma_i = 1 \\ 0 & \text{if } \sigma_i = 0. \end{cases} \tag{1}$$

where $A_{ij}$ are determined via the least-squares regression.

The differences between the actual and predicted values of the abundance of species $i$ across all possible communities can be used to compute the coefficient of determination $R_i^2$. The performance of the linear model in predicting the abundance of all species in the community is then characterized by the average of $R_i^2$ across all the species. This average $\overline{R^2}$ is close to zero when the linear model fails to fit the data; $\overline{R^2}$ is close to one when the assembly map is approximately linear and species abundances are explained well by independent contributions from other community members.

Fig 2 shows that community assembly becomes simpler as the number of interspecific interactions decreases. For low niche overlap, we obtained good linear fits and large values of $\overline{R^2}$. We can understand this in the limiting scenario, where each species consumes one, separate resource; in this scenario, species abundance is independent of presence of other species and instead depends only its consumption strength, resource supply, and mortality (see S1 Text Sec.5). In contrast, when niche overlap was high, linear fits were poor and $\overline{R^2}$ was small. This behavior was consistent across different numbers of candidate species and resources, so we chose the niche overlap as the default method to control community structure. We also checked whether our conclusions are robust by changing environmental complexity or by including metabolic cross-feeding; see the results below and S2 Fig.

## The role of community function

The optimization process could be affected not only by the interspecific interactions, but also by the community function that is being maximized. To investigate this possibility, we studied the efficacy of the search process for four choices of the community function described below. Since consumer-resource models simplify details of microbial metabolism, we did not explicitly model the production of a desired metabolite. Instead, we followed experimental studies in expressing the desired community function in terms of species abundances in the community [65, 66].

The first three community functions were chosen to cover scenarios ranging from functions dependent on the contributions of many species to functions dependent on just a single

species. The first community function (diversity) was the diversity of the community. We explored this function because it could be predictive of the overall community productivity in some ecosystems [67–70]. The precise definition of the diversity did not affect our conclusions (see Methods), so we focused on the Shannon Diversity (Eq 7). The second community function (pair productivity) was the product of the abundances of two specific microbes (Eq 9). This choice was motivated by the communities where the combined efforts of two species ware necessary to produce a metabolite. The third community function (focal abundance) was simply the abundance of a single species, which could reflect the production or degradation of a specific metabolite. The fourth function ('butyrate production') was inspired by the experimental study on gut microbes [65], which parameterized a model of butyrate production in terms of the species abundances in the final community (Eq 10). Even though our simulations do not explicitly describe butyrate production, we chose this nonlinear index of community productivity to account for the complexity of ecological functions important in applications and to test the robustness of our findings for simpler measures described above. To generate the landscapes, we randomly matched species in simulations and experiments to apply the inferred model on simulated data (see Methods). The random matching of species in simulations and experiments emphasizes that our goal was not to precisely reproduce experimental landscapes, but potentially emulate properties of the experimental landscape at some coarse level.

We quantified the search efficacy as the ratio of the best community function found by the search $\mathcal{F}_{found}$ to the highest value $\mathcal{F}_{max}$ across the entire landscape. To reduce the influence of the starting community, we report the search efficacy averaged over all possible starting communities.

Qualitatively, the results were the same for all community functions: The efficacy of search decreased with increasing niche overlap (Fig 3). We however found important quantitative differences. While optimization of diversity showed only a modest drop with higher niche overlap, the search efficacy for pair productivity plummeted to near zero. This sharp drop was due to sporadic coexistence of the two species. When the two species do not coexist in a region of the landscape, the community function is zero in that neighborhood and the search cannot progress. Search efficacy for butyrate production also dropped similarly because butyrate production increased when certain species pairs coexisted. This challenge can be partly overcome by starting search from the best out of many random communities instead of a single one, which we will discuss in a later section.

## Ruggedness of ecological landscapes

Niche overlap is only one of many mechanisms that can produce ecological landscapes of varying complexity. For example, cross-feeding could lead to additional interactions not captured by the consumption matrix. It would therefore be quite valuable to develop a mechanism-free metric of landscape complexity. In population genetics and optimization theory, such metrics are known as landscape ruggedness [71–73]. Rugged landscapes with many scattered peaks are much harder to navigate than smooth landscapes with a single peak. Many measures of ruggedness exist, and some of the most commonly used ones are described in S1 Text. In the main text, we limit the discussion to the three metrics that we found to be the most useful for microbial communities. These metrics are described below, and their mathematical definitions are provided in Methods. See Fig 4 for an example workflow of how to compute the ruggedness of an ecological landscape.

The roughness-slope ratio $r/s$ quantifies deviation from linearity of the landscape [73, 74]. Since a perfectly linear landscape is always optimizable by the heuristic greedy search (see

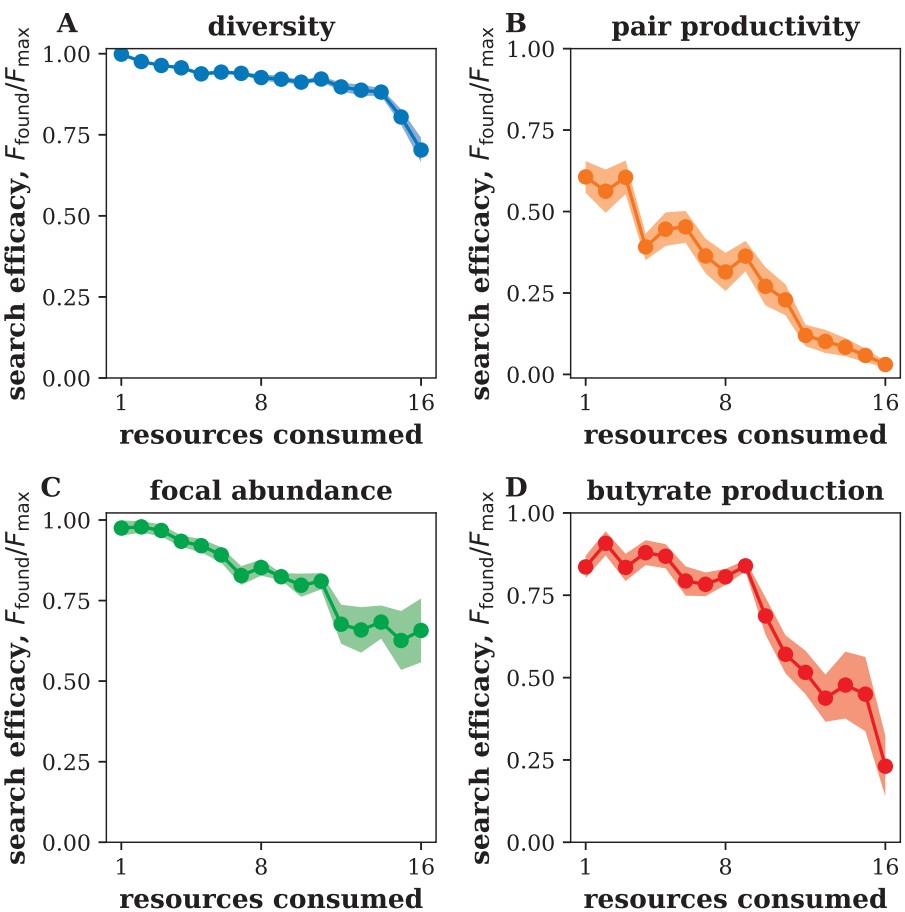

**Fig 3. Search efficacy decreases with niche overlap.** Search efficacy was quantified as the ratio of the best community function found by the search $\mathcal{F}_{\text{found}}$ to the highest value $\mathcal{F}_{\text{max}}$ across the entire landscape. The plots report the average and the standard error of the mean obtained from ten independently generated species pools. Different panels correspond to the four different community functions. The complexity of the landscape was controlled by varying the average number of resources that each species consumed $M_{\text{consumed}}$ while keeping the total number of resources $M_{\text{total}}$ fixed at $M_{\text{total}} = S = 16$. Data was from consumer-resource model simulations with 16 species shown in Fig 2. See Methods for a complete description of these simulations.

Methods), we expect search difficulty to increase with $r/s$. The roughness, $r$, measures the error of a linear fit to the community function landscape and the slope, $s$, measures the average magnitude of the change in $\mathcal{F}$ when a species is added or removed. Small values of $r/s$ indicate that there is a strong trend that is slightly obscured by local fluctuations. Such landscape are easy to navigate because the heuristic search can detect the direction of the gradient. The exact opposite occurs for large $r/s$ because large point-to-point fluctuations obscure the path towards the optimum.

A related, but different approach is to capture the changes in the community function upon adding or removing a single species. This metric is defined as $1 - Z_{\text{nn}}$, where $Z_{\text{nn}}$ is the average correlation between the values of $\mathcal{F}$ in communities that differ by adding or removing a single species. Large values of $1 - Z_{\text{nn}}$ indicate that changes in community function are uncorrelated and the search is difficult.

The third metric, $F_{\text{neut}}$, captures the challenge of quasi-neutral directions. If changes in $\sigma$ leave the community function nearly the same, then it is difficult to determine how to modify the community to increase $\mathcal{F}$. This situation occurs when the added species fails to establish or

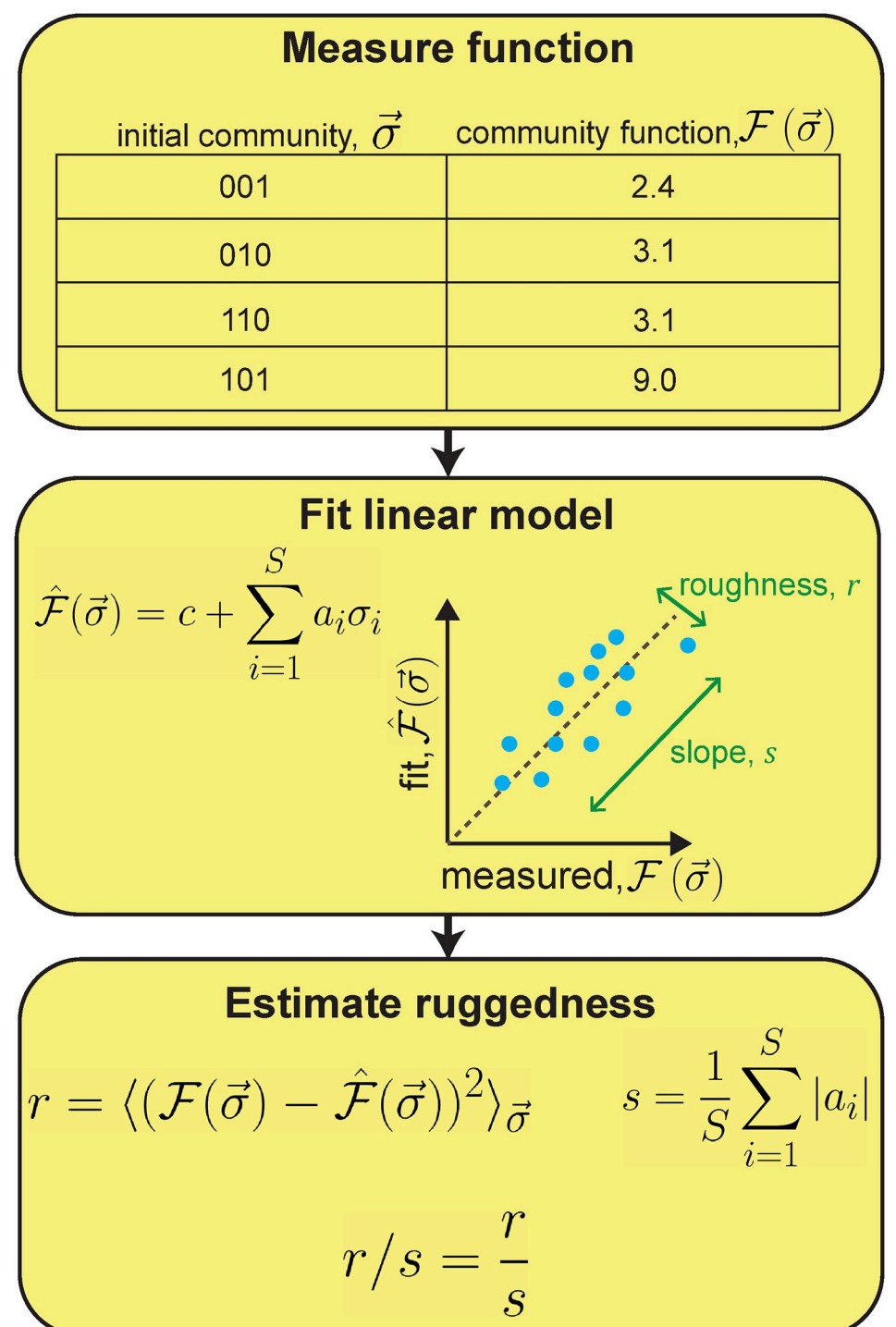

**Fig 4. Example of the workflow for calculating the ruggedness of ecological landscapes.** The roughness-slope ratio $r/s$ quantifies the deviation from linearity of the landscape. It is calculated by fitting the measured community function landscape with a linear model $\hat{\mathcal{F}}(\vec{\sigma})$ and then estimating $r$ and $s$ from the fit. The roughness, $r$, measures the average error of a linear fit to the community function landscape and the slope, $s$, measures the magnitude of the change in $\hat{\mathcal{F}}$. The calculation of other ruggedness metrics follows a similar pattern, but with different mathematical quantities being computed.

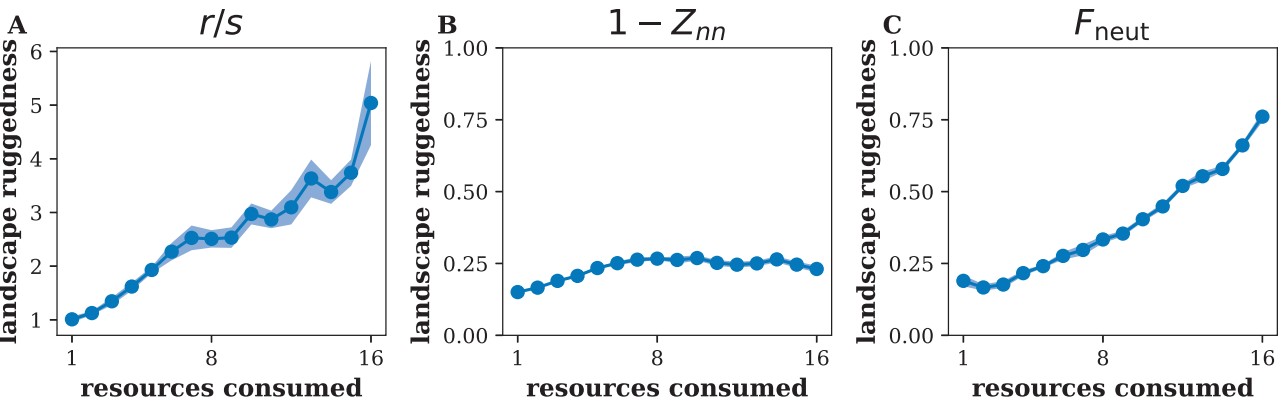

**Fig 5. Ruggedness of community function landscapes.** All metrics of ruggedness increased with niche overlap. Different panels correspond to different metrics of ruggedness. The plots report the average and the standard error of the mean obtained from ten independently generated species pools. The complexity of the landscape was controlled by varying the average number of resources that each species consumed, $M_{\text{consumed}}$, while keeping the total number of resources, $M_{\text{total}}$, fixed at $M_{\text{total}} = S = 16$. See Methods for the complete description of these simulations. The community function here was community diversity; see S3 Fig for other community functions.

fail to affect the function of interest despite invading. Quasi-neutral directions are common in some landscapes and are known to be a major impediment to optimization [75–77]. Thus, the fraction of quasi-neutral directions $F_{\text{neut}}$ could be a valuable predictor of the search success.

We analyzed how these three metrics change with the niche overlap across different community functions. In all cases, we found that landscape ruggedness increases with the number of interspecific interactions. The plots in Fig 5 are representative of the behavior that we observed. Note that the magnitude and pace of increase varied among the metrics, for e.g., $1 - Z_{\text{nn}}$ appears to increase and then saturate. Such differences are expected because metrics quantify different aspects of landscape structure, and they suggest there could be important differences in their ability to predict the efficacy of the heuristic search.

We also examined potential limitations of ruggedness metrics. Because they map the entire ecological landscape into a single number, ruggedness measures may not fully capture the actual structure of the search space. Hence, we looked for a ruggedness metric that captures the landscape with more than just a single number. Moreover, we sought to capture nonlinear effects, which are typically ignored by the commonly-used measures of ruggedness. To capture higher-order effects, we examined the fits of $\mathcal{F}(\vec{\sigma})$ by models of increasing complexity. Starting with the linear model, we added quadratic, cubic, quartic, etc terms in $\vec{\sigma}$ (see Methods). These terms account for non-pairwise interactions, e.g. interactions influenced by other species. Such conditional interactions could occur via a variety of mechanisms; for example, a species can induce an interaction between two other species by producing a nutrient that is consumed by both of these other species.

The model errors (unexplained variance) decreased approximately exponentially with the model complexity (Fig 6). The rate of decrease was similar for smooth and rugged landscapes with rugged landscapes always requiring a more complex model to achieve similar accuracy. We found no compelling examples to justify characterizing landscape ruggedness by more than a single number. Therefore, we focused on the simple metrics of landscape ruggedness defined above.

## Search is harder on rugged landscapes

Based on the above intuition, we expect search to be more difficult on rugged landscapes. However this relationship still needs to be tested and quantified in order to determine which,

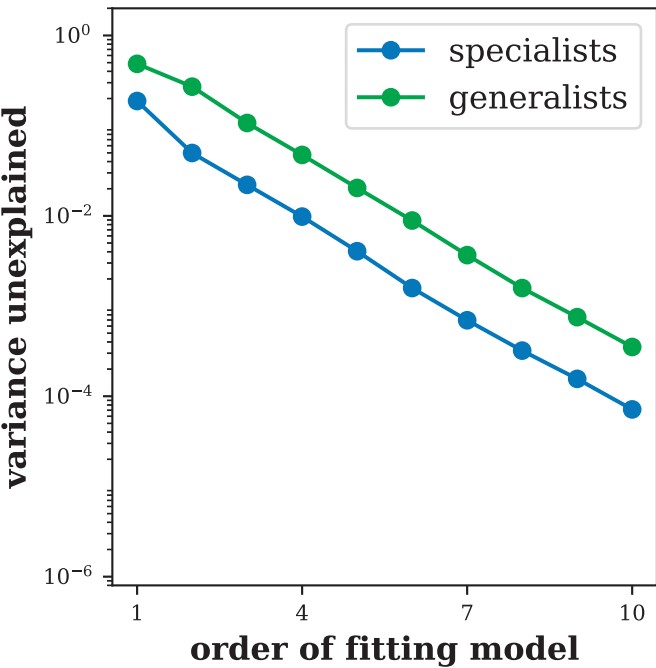

**Fig 6. Contribution of higher-order interactions.** Including higher order terms in the fitting model reduced the variance in the data not explained by the fit. First order fitting models are linear, second—quadratic, etc. The decrease in unexplained variance was approximately exponential. We used $S = 16$, $M_{consumed} = 2$ for specialists and $M_{consumed} = 14$ for generalists; the community function was Shannon diversity. All other simulation parameters are the same as in Fig 5.

if any, of the ruggedness measures can be used in real-world applications. So, we evaluated the search efficacy and ruggedness for the landscapes from Fig 5. All three ruggedness metrics were significantly anti-correlated with the search efficacy ($p \ll 10^{-4}$).

The examples of these correlations are shown see Fig 7, which also illustrates how the strength of the correlation varies with the metric and community function. Notably, the roughness-slope ratio consistently showed the strongest anti-correlation, and ruggedness measures were more predictive of functions determined by groups of interacting species. Overall, we found that landscape ruggedness is a good predictor of the search success across all situations examined.

## Landscape ruggedness can be estimated from limited data

So far, all of our results were obtained with the full knowledge of the ecological landscape. In practical applications, however, one can examine only a very limited set of species combinations. Thus, it is important to determine whether the ruggedness of a landscape can be determined from limited data.

We estimated different ruggedness metrics for various fractions of the complete landscapes. This was done differently for different metrics because some of them require information on the nearest neighbors while others do not; see Methods. The results showed that ruggedness can indeed be well estimated from as little as 0.1% of the total number of possible communities (Fig 8A and 8B). Moreover, the estimated ruggedness remained highly informative of search success (Fig 8C). The practical utility of ruggedness estimates was further confirmed by their robustness to the measurement noise (S4 Fig).

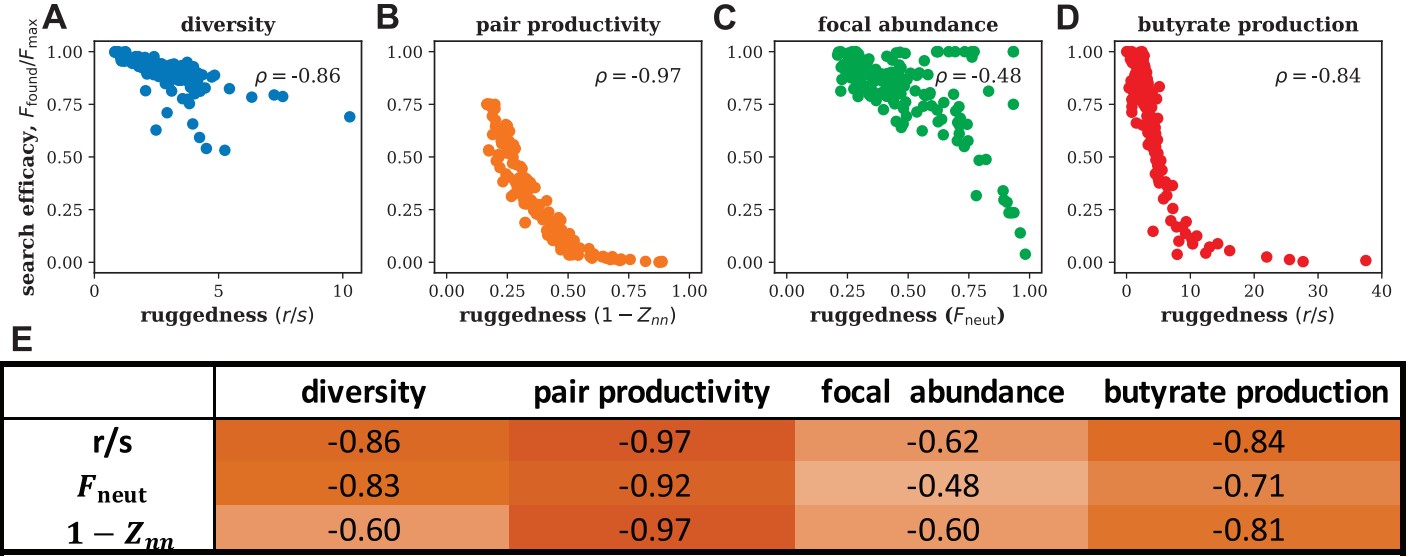

**Fig 7. Landscape ruggedness inhibits search success.** Search efficacy is anti-correlated with the landscape ruggedness. Panels (A), (B), (C), (D) show examples for different community functions and ruggedness metrics. Each point corresponds to a separate $S = 16$ species pool studied in Fig 5. The systematic effects of the ruggedness metric and the type of community function is tabulated in (E), which reports the Spearman's correlation coefficient $\rho$ between search efficacy and ruggedness. The correlations were highly significant ($p \ll 10^{-4}$) in all cases. Darkness of table background color indicates magnitude of correlation.

### Results generalize to other search protocols and the presence of cross-feeding

We further tested the utility of ruggedness measures by applying them to different models and search protocols.

The standard consumer-resource models are undeniably simpler than real microbial communities and have certain mathematical properties that may not hold in more general settings [41, 44]. Since metabolic exchanges are central to most applications [34, 35, 78], we tested the

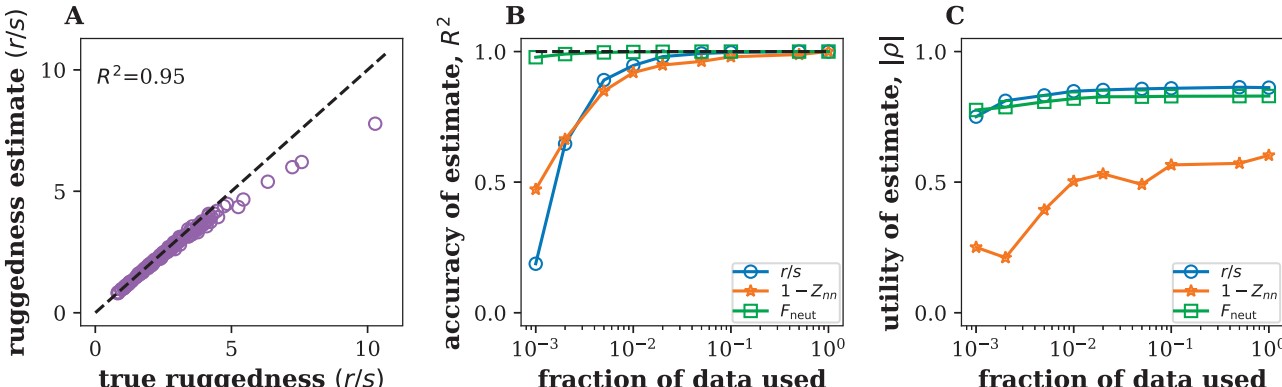

**Fig 8. Estimating ruggedness from limited data.** (A) True and estimated ruggedness are tightly correlated when only 1% of the data is available for the estimation. The data are from Fig 5, and each point corresponds to a landscape with a different degree of niche overlap. (B) The accuracy of the estimation improves with the amount of available data for all ruggedness measures. (C) The estimated ruggedness remain highly informative of search efficacy. The utility of the ruggedness estimate, $|\rho|$, is measured as the magnitude of the Spearman's correlation coefficient between search efficacy and estimated ruggedness. Note that $|\rho|$ remained close to one even for very low fractions of the data for which $R^2$ in panel B started to decrease rapidly suggesting that deviations between predicted and actual roughness (see A and B) do not impede the prediction of search success. For this figure, the community function is the Shannon diversity. Similar results were obtained for other functions tested.

 

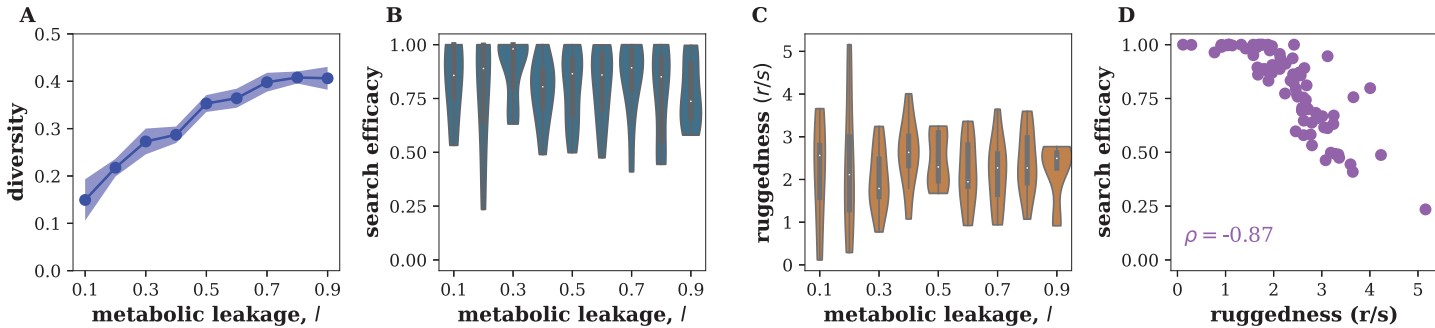

**Fig 9. Ruggedness is predictive of search efficacy in models with cross-feeding.** Cross-feeding altered the composition of microbial communities (A), but had no systematic effect on the success of the heuristic search (B) or ruggedness (C). Importantly, the search efficacy and ruggedness remained strongly anti-correlated in communities with cross-feeding (D). Simulated species pools had $S = 12$ species that leaked a fraction $l$ of the resources they consumed as consumable byproducts. Community function used in panels B, C, D is the abundance of a focal species. A strong negative anti-correlation was found for other choices of community function as well (See S6 Fig). Only one out of 12 modeled resources was supplied externally; the rest were present only due to metabolic leakage. Each species was able to consume 6 resources. Ruggedness was quantified by $r/s$. See Methods for further simulation details.

robustness of our results by adding cross-feeding into our model. This was accomplished by specifying the "metabolic leakage matrix" that linked the consumption of each resource to the production of another resource. A certain fraction, $l$, of the produced resources was allowed to "leak" into the environment, where it could be consumed by all species [33].

First, we checked whether our results still hold when there is a nontrivial amount of leakage. This was indeed the case, and landscape ruggedness was predictive of the search efficacy when we varied the number of resources consumed; see S5 Fig. Then, we simulated communities supplied with a single resource at varying amounts of leakage to examine how leakage affected community composition, landscape ruggedness, and search efficacy. Leakage tended to increase the number of surviving species and the diversity of the steady-state communities (Fig 9A). However, it had no systematic affect on any of the ruggedness measures or search efficacy (Fig 9A and 9C). Importantly, when we pooled the simulations across all leakage levels, we still found a strong negative relationship between landscape ruggedness and the success of the search (Fig 9D). We found a similarly strong negative relationship for other choices of community function (S6 Fig) and also when number of supplied resources was varied (S7 Fig).

The choice of the search protocol also had no major effect on our conclusions. We considered three variations of the simple gradient-ascent considered so far. The first variation was a simple improvement: instead of starting from a single community, the search commenced from the best of $q$ randomly chosen communities. This procedure helped reduce variability of the search length and avoid really poor outcomes due to an unfortunate starting point. We denote this search protocol 'one-step reassembly' as communities at each iteration need to be assembled from scratch using some of the $S$ species.

We also considered two other protocols that mirrored recently proposed community selection methods [79]. These protocols modified the existing community instead of reassembling it from isolates. While adding another species to an existing community is straightforward, removing a species is not. A typical solution of this problem is to perform a strong dilution such that the one or more species become extinct due to the randomness of the dilution process. Our second search protocol was to start from the best of the $q$ communities and then dilute it to create $q$ new communities and $q-1$ of these new communities are diluted further to promote stochastic extinctions. Then then best community is selected for the next iteration. The third search protocol was the same as the second, except a different random species was

 

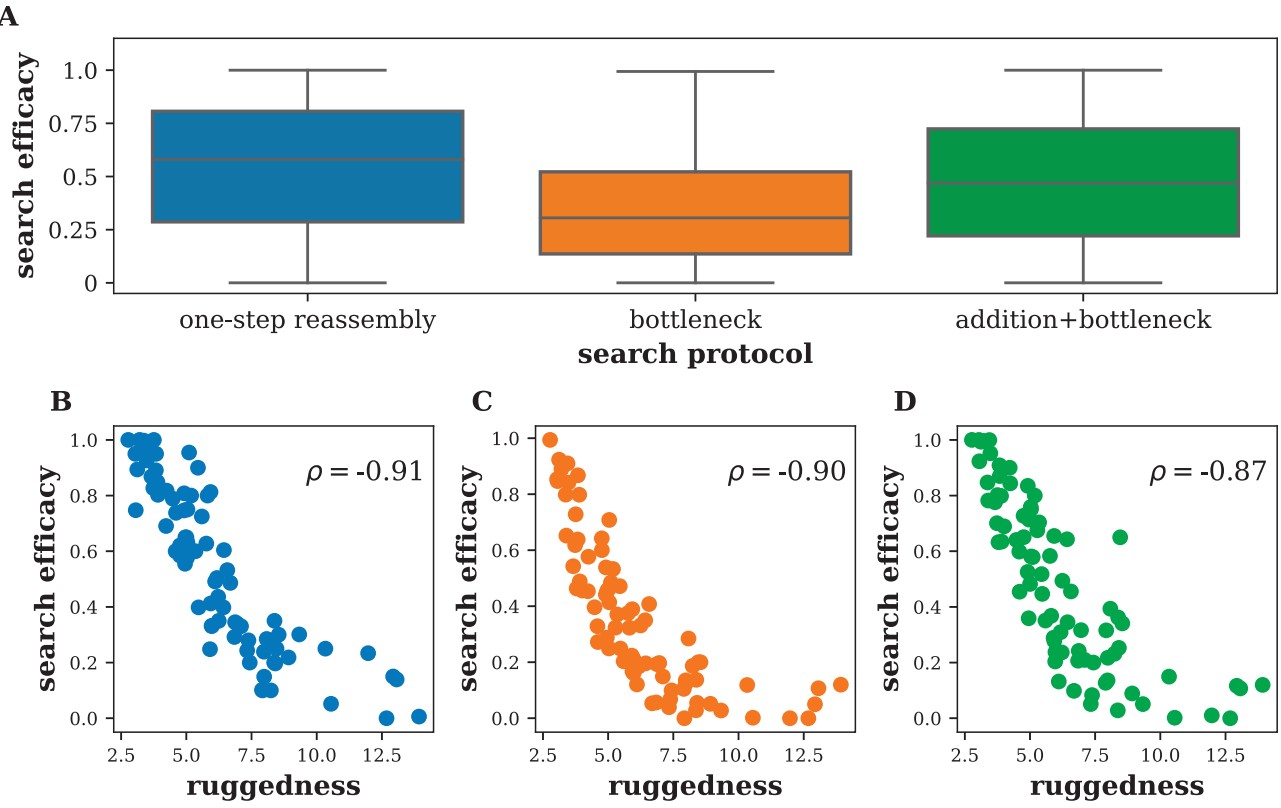

**Fig 10. Ruggedness is informative for a variety of search protocols.** (A) shows the search efficacy of three different community optimization protocols. Despite differences in the performance of different protocols, ruggedness remained informative of search success across protocols (panels B, C, D). Community function was pair-productivity. All searches started from the best of $q = 13$ communities. Dilution rate in the dilution-based protocols was adjusted for the highest performance. Simulated communities were the same as in Fig 9.

added to each of the $q-1$ strongly diluted communities. We denote these two dilution-based search protocols as 'bottleneck' and 'addition + bottleneck' respectively.

While the dilution protocols could be appealing, it is worth noting that their performance strongly depended on the dilution factors, which needed to be carefully adjusted for optimal performance. Outside of this narrow range, either too many species went extinct or all species survived (S8 Fig). For each protocol, we chose the dilution rate that yielded the best performance. In other words, our results are for the best-case scenario.

The search efficacy differed substantially among protocols. The 'one-step reassembly' and 'addition + bottleneck' performed best in our tests. Importantly, the difference in performance did not affect the relationship between landscape ruggedness and search efficacy (Fig 10B, 10C and 10D).

## Ruggedness of experimental landscapes

We also were able to examine the relationship between landscape ruggedness and search efficacy in an experimental data set. Unfortunately, we found only a single study that had a number of complete ecological landscapes. Langenheder, et al. [32] measured the metabolic activity of a community of six soil microbes grown on seven different carbon sources in all possible species combinations (Fig 11).

To check whether these landscape are similar to the ones from consumer-resource models, we determined how well they can be fitted by models of varying complexity (compare to Fig

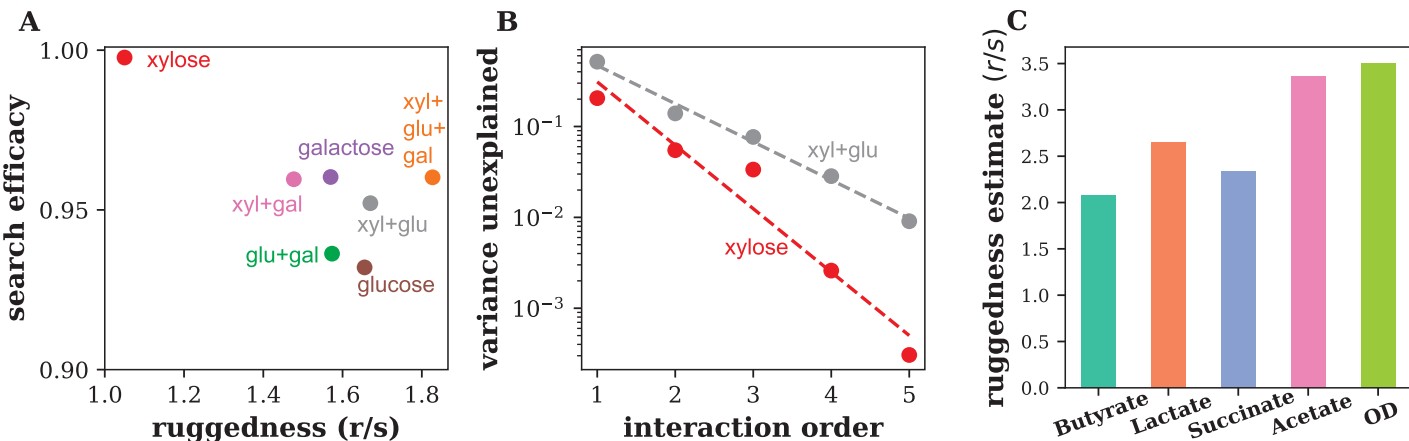

**Fig 11. Ruggedness of experimental landscapes.** (A) shows the ruggedness and search efficacy in the seven environments studied in Ref. [32] labeled by the carbon sources used. There is an apparent anticorrelation between the efficacy and ruggedness, but it does not reach statistical significance ($p = 0.29$). Note that the range of ruggedness and search efficacy is quite narrow presumably because of the overlap in the carbon sources in different environments and the choice of species that grow well on these sugars. Nevertheless, the search efficacy was highest on the least rugged landscape (xylose). Search efficacy and ruggedness were anti-correlated with ruggedness on the three pure carbon sources as well. (B) The decrease in the variance unexplained with increasing model complexity shows behavior similar to that of the simulated landscapes in Fig 6. Dashed lines in panel B show a linear fit on the semi-logarithmic plot.(C) The estimated ruggedness of experimental landscapes of five different community functions studied in Ref. [65]. The ruggedness was estimated of the 25 species landscape was estimated from experimental measurements of community function of 577 different species combinations.

6). For all seven experimental landscapes, we found that the accuracy of the fit dramatically improved the complexity of the model and that the quantitative metrics of ruggedness were comparable to those in simulated consumer-resource models (Fig 11B).

Encouraged by these observations, we computed landscape ruggedness and estimated the performance of the heuristic search. The results are shown in Fig 11A. In agreement with our expectations, we found that the ruggedness and efficacy were anti-correlated (Spearman's correlation coefficient, $\rho = -0.46$), but this anti-correlation did not reach statistical significance ($p = 0.29$). The lack of significance could be attributed in part due to the small size of the data set and in part due to the narrow range of observed search efficacies. This latter factor might be due to the overlap in the carbon sources in different environments and the choice of species that grow well on the carbon sources in monoculture. Notably, the search efficacy was the highest on the least rugged landscape (xylose).

Other surveys of microbial community might reflect the complexity of applications better, but they do not sample all possible communities. Nevertheless, such surveys could allow us to estimate ruggedness measures and make predictions about the efficacy of the search, which can then be tested experimentally. To that end, we analyzed data from Clark, et al. [65]. This study examined communities formed by 577 different combinations of 25 different prevalent gut bacterial species. It measured the four different organic acid fermentation products (butyrate, lactate, succinate, and acetate) as well as the optical density (OD) of each community. We estimated the ruggedness of the landscapes formed by organic acid production and OD from the experimental measurements (Fig 11C). Our analysis indicates that butyrate production, which the study aimed to optimize was the least rugged of the five landscapes assayed. Hence, heuristic greedy search may be a promising experimental strategy for optimizing butyrate production.

## Discussion

Many problems in biotechnology can be solved by a well-chosen microbial community. However, designing communities is a challenging and multifaceted process. At the very least, it involves identifying promising candidate species and then sifting through the combination of these species to find a productive and stable community. Since an exhaustive assessment of all possible species combinations is rarely possible, an efficient search protocol is key to the design of microbial communities.

A number of such search protocols have been proposed so far [14, 15, 80, 81]. All of these protocols share the idea of selecting the best performing communities at each stage of the search. However, many of them failed to improve on the best community in the starting pool because they did not modify the selected communities between search stages [79]. Search protocols that modify the selected community at each search stage should fare better. Protocols based on reassembling the community at each stage, like the ones examined here, are a promising way of systematically exploring the space of possible communities.

To uncover the factors controlling the success of search protocols, we introduced the concept of a community function landscape that describes how the community function changes with community composition, in the context of community design. We focused on ecosystems in which the population comes to a unique steady state that is fully determined by the presence-absence of microbes in the starting community. Although such simple dynamics appear to be quite common, there are well-known exceptions (Table 1). Often, these special cases occur only in a limited region of the parameter space and, at a practical level, can be ignored. This simplification allowed us to rigorously define the community function landscape and propose simple measures to characterize its structure.

We found that the success of a heuristic search largely depends on the metrics of landscape ruggedness. Three ruggedness metrics stood out as the most useful: $r/s$, $1 - Z_{nn}$, and $F_{neut}$. The roughness-slope metric, $r/s$, was the most predictive in our studies, and estimates from limited data remained highly informative of search. The fraction of neutral directions, $F_{neut}$, was also predictive and easy to estimate, but had an important downside: It required a quantitative threshold for distinguishing neutral directions, which cannot be determined without additional experiments. In practice, the best metric may vary depending on the nature of the ecosystem and experimental constraints.

We also identified how landscape ruggedness depends on the ecology of the microbial community. Our simulations showed that substantial overlap in the metabolic niches of the species increase the ruggedness of the landscape and reduce the search efficacy. Thus, it could be important to choose the species and environmental variables in a manner that minimizes unnecessary interactions.

Our model focused on competition for resources and ignored other processes such as pH-mediated interactions or chemical warfare. While such processes certainly play an important role in some ecosystems, they are not always present, especially in engineered communities. In fact, a recent study of reconstituted human gut microbiota showed that resource competition explained majority of the interspecific interactions [30]. Nevertheless, the effect of non-metabolic interactions on ecological landscape is worth further investigation, but this research would require a well-characterized set of modeling approaches akin to the consumer-resource models used in our work.

We also neglected some important aspects of metabolic interactions such as sequential consumption of resources. While switching among different resources is widespread in nature, it mostly affects the initial dynamics in batch culture experiments, where preferred resources are consumed first, and the species then switch to other resources. For chemostat-like dynamics

considered here, once the community comes to a steady state, no diauxic shifts occur, and one can view the consumption rates in our simulations as realized consumption rates.

While this study relied heavily on consumer-resource models, we expect that our conclusions hold more generally. Indeed, all of our results remained unchanged when various levels of cross-feeding was incorporated in the model. The analysis of experimental landscapes corroborated our conclusions as well.

Navigating complex landscapes is a long-standing problem in physics, mathematics, and computer science, and no algorithm is guaranteed to succeed in all optimization problems [82–85]. Nevertheless, the ecological landscape framework and the ruggedness measures developed in this study could help develop and analyze better search algorithms. For example, a greedy search with larger step sizes might improve performance on some landscapes. In our context, this can be accomplished by adding or removing multiple species simultaneously. The choice of step size could be informed by $Z_{nn}$ (related to correlation length of landscape) and landscape neutrality $F_{neut}$ (see [86–88]). In landscapes of pair productivity, a large portion of the landscape was neutral because community function was zero when one of the partners did not survive; we found that repeating the search from many starting points can improve search outcomes on these landscape (compare Figs 7 and 10). One could also adapt well-studied optimization algorithms such as as branch and bound, simulated annealing, or stochastic gradient descent [82, 83, 89], but these algorithms may require more search steps than experimentally feasible. Landscape properties, such as those measured by various ruggedness metrics, could be used to identify the best search algorithms for a given landscape (see [90, 91]). Exploring the large space of possible search strategies, with ruggedness characterizing landscape structure and serving as a benchmark of problem difficulty, is an exciting direction for future research.

Ecological landscapes are very versatile because they can be inferred from both simulations and experiments and can even be compared across different ecosystems. Furthermore, landscapes can be reconstructed simply by measuring the community function across different species combinations. In contrast, alternative optimization approaches require sequencing to infer microbial abundances and a mathematical model to predict desired community composition [65, 92–94] or focus on communities with just two species [95–97]. Ecological landscapes may also facilitate the diffusion of knowledge from other disciplines. One such example is the idea of community coalescence that is analogous to recombination in population genetics [98]. In sum, the versatility of ecological landscapes and the practicality of ruggedness metrics can make them a valuable tool for bioengineering applications.

## Methods

### Consumer-resource models with and without cross-feeding

We simulated a consumer-resource model where species competed for $M_{total}$ resources, which were supplied externally [42, 99]. A species $i$ had abundance $n_i$, growth rate $g_i$, and maintenance cost $m_i$. A resource $\alpha$ had concentration $R_\alpha$ and quality $w_\alpha$. Resource supply resembled a chemostat with supply point $R_\alpha^0$ and dilution rate $\tau^{-1}$. Species differed in their preferences for the various resources, encoded in the consumption preference matrix $C$. A species could consume $M_{consumed}$ randomly chosen resources out of the $M_{total}$ supplied resources. The strength of this preference was randomly drawn from a gamma distribution. The equations governing

the dynamics of the species and resources are:

$$\frac{dn_i}{dt} = g_i n_i \left[ \sum_{\alpha=1}^{M} w_\alpha \mu(C_{i\alpha} R_\alpha) - m_i \right], \qquad (2)$$

$$\frac{dR_\alpha}{dt} = \tau^{-1}(R_\alpha^0 - R_\alpha) - \sum_{i=1}^{S} n_i \mu(C_{i\alpha} R_\alpha), \qquad (3)$$

where $\mu$ is a function describing resource uptake rates. For the consumer resource model without cross-feeding, we assumed a simple linear form for the uptake.

Simulations were carried out in python using custom code and a modified version of the community simulator package [37]. Obtaining the steady state via direct numerical integration of the ODEs is computationally expensive as the number of species combinations grows exponentially with pool size. Therefore, we utilized a recently discovered mapping between ecological models and convex optimization to calculate the steady state quickly [37, 100]. Species with an abundance below $10^{-6}$ were set to be extinct.

To incorporate the secretion and uptake of metabolites by microbes, we examined the microbial consumer-resource model with cross-feeding [33]. In the cross-feeding model, each species leaks a fraction, $l$, of resources it consumes in the form of metabolic byproducts. The composition of these byproducts is specified by the metabolic leakage matrix $\mathcal{L}$, with matrix element $\mathcal{L}_{\alpha\beta}$ specifying the amount of resource $\beta$ leaked when the species consumes resource $\alpha$. This leakage is weighted by the ratio of resource qualities $w_\beta/w_\alpha$ so that energy-poor resources cannot produce a disproportionate amount of energy-rich resources. Each row of the leakage matrix sums to one. The leakage matrix and resource qualities were independent of species identity to respect potential universal stoichiometric constraints on species metabolism. The dynamical equations of the model with cross-feeding are:

$$\frac{dn_i}{dt} = g_i n_i \left[ (1-l) \sum_{\alpha=1}^{M} w_\alpha \mu(C_{i\alpha} R_\alpha) - m_i \right],$$

$$\frac{dR_\alpha}{dt} = \tau^{-1}(R_\alpha^0 - R_\alpha) - \sum_{i=1}^{S} n_i \mu(C_{i\alpha} R_\alpha) + l \sum_{i=1}^{S} \sum_{\beta=1}^{M} n_i \mathcal{L}_{\alpha\beta} \frac{w_\beta}{w_\alpha} \mu(c_{i\beta} R_\beta). \qquad (4)$$

Here, resource uptake rates were assumed to be of Monod form, $\mu(x) = x/\left(1 + \frac{x}{K_\alpha}\right)$. We simulated the system of ODEs explicitly using the community simulator package to obtain steady-state abundances. Since direct simulation of the ODEs is computationally expensive, we simulated smaller candidate species pools with $S = 12$. Simulations reached steady state if the root mean square of the logarithmic species growth rates rates fell below a threshold, i.e., $\text{RMS}\left(\frac{1}{n_i} \frac{dn_i}{dt}\right) < 10^{-3}$. We imposed an abundance cutoff at periodic intervals during the numerical integration to hasten the extinction of species. We verified that the extinct species could not have survived in the community by simulating a re-invasion attempt of the steady-state community.

## Simulation parameters

In simulations without cross-feeding, we simulated candidate species pools with $S$ species and supplied $M_{\text{total}}$ resources. All species within a candidate species pool consumed a fixed number of resources, $M_{\text{consumed}}$, chosen randomly for each species. The strength of these non-zero

consumption preferences were chosen from a gamma distribution, motivated by previous work [41], with parameters $(k, \theta) = (10, 0.1)$. Growth rates $g_i$ and death rates $m_i$ were sampled from normal distributions with mean 1.0 and variance 0.1(truncated to ensure positivity). Resource quality $w_\alpha$ and chemostat fixed point $R_\alpha^0$ were sampled from gamma distributions parameterized by $(k, \theta) = (10, 0.1)$ and $(k, \theta) = (10, 2)$; chemostat dilution rate $\tau^{-1}$ was set to 1.

In these simulations, we varied $M_{\text{consumed}}$ from 1 resource to $M_{\text{total}}$ resources across different candidate pools. We simulated ten pools at each value of $M_{\text{consumed}}$. We examined candidate pools of varying sizes: Fig 2 corresponds to simulations for $(S, M_{\text{total}}) = \{(4, 4), (8, 8), (12, 12), (16, 16)\}$. Figs 3, 5, 6, 7, and 8 present data from the 16 species candidate pools. We also simulated and analyzed results for $(S, M_{\text{total}}) = \{(8, 4), (12, 6), (16, 8)\}$ and $(S, M_{\text{total}}) = \{(8, 16), (12, 24), (16, 32)\}$ to verify that our results are robust to the number of supplied resources (S2 Fig).

In simulations with cross-feeding (Figs 9 and 10), we simulated $S = 12$ candidate species and $M_{\text{total}} = 12$ resources. Only a single resource, $R_0$, was supplied externally. The resource was supplied in sufficient amounts (with a chemostat fixed point $R_0^0 = 240$) that many species were able to survive by feeding on the byproducts of the consumers of the supplied resource. Species leaked a fraction $l$ of the consumed resources in forms specified by the leakage matrix. Each column of the leakage matrix was sampled from a Dirichlet distribution with parameter 0.5. Using the Dirichlet distribution guaranteed that each column summed to one. The Monod form for resource uptake had parameter $K_\alpha = 20$. The chemostat dilution rate was set to 0.1. All other parameters: growth rates $g_i$, death rates $m_i$, resource quality $w_\alpha$, and strength of consumption preferences $C_{i\alpha}$, were sampled as in the consumer-resource model.

We performed three types of simulations with the cross-feeding model—examining the effect of the extent of cross-feeding, environmental complexity, and niche overlap. Parameters were chosen as below:

1. To examine the effect of the extent of cross-feeding, we varied the metabolic leakage fraction $l$ from 0.1 to 0.9 across species pools. The number of resources consumed by each species, $M_{\text{consumed}}$, was 6. To ensure that at least one species in every candidate pool could consume the supplied resource, we constrained species 0 to always consume the sole supplied resource $R_0$ and species 1 to never consume $R_0$. (Both species could still only consume 6 resources in total). The abundance cutoff was $10^{-6}$. These simulations are analyzed in Figs 9 and 10.

2. To examine the impact of environmental complexity, we varied the number of resources supplied, $R_{\text{supplied}}$, from 1 to 12 across species pools. We fixed $l = 0.5$ and $M_{\text{consumed}} = 6$. We fixed the total supply flux of resources by setting the individual resource supply points to $240/R_{\text{supplied}}$. The abundance cutoff was $10^{-3}$. Results are shown in S7 Fig. Similar results were obtained when we allowed the total resource supply flux to increase with the number of supplied resources (each individual resource had supply point at 240).

3. To examine effect of niche overlap, we vary the number of resources consumed from 1 (extreme specialists) to 12 (extreme generalists). The metabolic leakage fraction was fixed at $l = 0.5$. We constrained species 0 to always consume the sole supplied resource $R_0$, to ensure that at least one species ate the supplied resource. The abundance cutoff was $10^{-3}$. Results are shown in S5 Fig.

For each set of parameters, we generated 10 candidate pools differing only by the random sampling.

## Characterizing the map to steady-state abundances $\vec{N}(\vec{\sigma})$

Using the simulation models parameterized as above, we simulated all possible combinations of species in the candidate species pools till they reached steady-state. To characterize the assembly map $\vec{N}(\vec{\sigma})$, we quantified how well observed abundance of each species is explained by adding independent contributions from other community members. Therefore, we fit the steady-state abundance of a species $i$, $N_i(\vec{\sigma})$, using the model

$$N_i(\vec{\sigma}) = \begin{cases} \sum_{j=1}^{S} A_{ij}\sigma_j + \epsilon, & \text{if } \sigma_i = 1 \\ 0 & \text{if } \sigma_i = 0. \end{cases} \tag{5}$$

where $\sigma_i$ and $\sigma_j$ denote the presence-absence of species $i$ and $j$, $A$ is the matrix of effective interactions, and $\epsilon$ is the model error. Note that the model can predict negative abundances, which we set to zero. This truncation did not affect our results (see S9 Fig).

For each species $i$, we perform a least squares linear regression to infer the corresponding row (row $i$) of matrix $A$ using data from communities where the species was initially present ($\sigma_i = 1$).

The abundance predicted by the model, $\hat{N}_i(\vec{\sigma})$, is given by

$$\hat{N}_i(\vec{\sigma}) = \begin{cases} \sum_{j=1}^{S} A_{ij}\sigma_j & \text{if } \sigma_i = 1 \\ 0 & \text{if } \sigma_i = 0. \end{cases} \tag{6}$$

We record the fraction of the variance explained (equivalent to the coefficient of determination) by the model, and repeat the regression for each of the $S$ species. The average variance explained across all $S$ species, $\overline{R^2}$, is a measure of the linearity of the abundance landscape. Results are presented in Fig 2.

## Community function landscapes

We computed the community function of interest, $\mathcal{F}$, from the steady-state species abundances. The three forms of community functions we analyzed are:

1. Diversity was measured as Shannon Diversity, defined by

$$\text{diversity} = \frac{1}{\ln S}\sum_{i=1}^{S} p_i \ln p_i, \tag{7}$$

where $p_i$ is the relative abundance of species $i$. Note that we had also considered an alternative measure of diversity, Simpson's diversity, which we do not discuss for brevity since results were similar. Simpson's diversity was defined as

$$\text{Simpsons Div.} = \sum_{i=1}^{S} 1 - p_i^2. \tag{8}$$

We do not discuss this function for brevity since results were similar (correlation of search efficacy with $r/s$, $1 - Z_{nn}$, and $F_{neut}$ on Simpson's diversity landscape was -0.83, -0.83, and -0.35.

2. Pair Productivity, mediated by a pair of species $(A, B)$, was measured as the product of their abundances,

$$\text{pair productivity} = N_A N_B. \tag{9}$$

This approximates a scenario where a useful product is synthesized by the combined activity

of two species. Results are shown for the species pair with indices 2, 3 in simulations (randomly chosen).

3. Focal abundance. The abundance of a focal species of interest was given by $N_{\text{focal}}$. This can also be understood as being a component of the vector $\vec{N}(\vec{\sigma})$. Since this function makes sense only if the species was present in the initial community, we restricted our analyses to the subset of the complete landscape where the focal species was initially present. This subset resembles a landscape made by $S - 1$ species, with $2^{S-1}$ species combinations.

4. Butyrate production. To simulate butyrate production, we used the model inferred in the study by Clark, et al. [65]. The study measured butyrate production of different combinations of 25 gut microbes to infer parameters in a mathematical model of butyrate production via L1-regularized regression. In addition to a dependence on the microbial abundance, butyrate production was also allowed to depend on the presence of species in the final community. This allowed for the possibility of species at small abundances having large effects on community function without being penalized by the regularization. The inferred model was of the form:

$$\text{butyrate prod.} = \sum_i (w_i^{(0)} \sigma_i^{\text{final}} + w_i^{(1)} N_i) + \\ \sum_{i,j} (w_{ij}^{(2)} \sigma_i^{\text{final}} \sigma_j^{\text{final}} + w_{ij}^{(3)} \sigma_i^{\text{final}} N_j + w_{ij}^{(4)} N_i N_j). \tag{10}$$

To emulate butyrate production in our simulated communities, we applied the model with coefficients measured in the study (model 'M3' in [65]) on our simulation results. In the model 'M3', 11 different species contributed significantly to the function. We assigned species $1 - 11$ in our simulations to one of the 11 experimental species at random. Then we used the inferred model parameters as provided in Ref. [65] to generate landscapes of butyrate production. The model 'M3' had 5 nonzero coefficients at first order ($w^{(0)}$ and $w^{(1)}$) and 13 nonzero coefficients of second order ($w^{(0)}$, $w^{(1)}$, and $w^{(2)}$).

## Landscape ruggedness

We used three ruggedness measures to characterize the community function landscapes:

1. The roughness-slope ratio $r/s$ is given by the ratio of roughness $r$ to slope $s$ of the landscape. The roughness, $r$, measures the root-mean-squared residual of the linear fit. It is obtained by fitting the community function $\mathcal{F}$ with the linear model, $\hat{\mathcal{F}}(\vec{\sigma}) = c + \sum_{i=1}^{S} a_i \sigma_i$, via least squares regression. $r$ is given by

$$r = \sqrt{\frac{1}{\|\vec{\sigma}\|} \sum_{\vec{\sigma}} (\mathcal{F}(\vec{\sigma}) - \hat{\mathcal{F}}(\vec{\sigma}))^2}, \tag{11}$$

where $\|\vec{\sigma}\|$ is the number of points on the landscape.

The slope, $s$, measures the magnitude of change in $\mathcal{F}$ when a species is added or removed. It is given by

$$s = \frac{1}{S} \sum_{i=1}^{S} |a_i|. \tag{12}$$

2. $1 - Z_{nn}$, where $Z_{nn}$ is nearest neighbor correlation on the landscape. $Z_{nn}$ is defined as:

$$Z_{nn} = \frac{\frac{1}{\|\vec{\sigma}^{nn}\|} \sum_{\vec{\sigma}, \vec{\sigma}^{nn}} \mathcal{F}(\vec{\sigma}) \mathcal{F}(\vec{\sigma}^{nn}) - \overline{\mathcal{F}}^2}{\sum_{\vec{\sigma}} (\mathcal{F}(\vec{\sigma}) - \overline{\mathcal{F}})^2}. \tag{13}$$

$\vec{\sigma}^{nn}$ refers to the nearest neighbors of $\vec{\sigma}$ on the landscape, i.e., communities obtained by adding or removing a single species, $\|\vec{\sigma}^{nn}\|$ is the number of nearest neighbors, and $\overline{\mathcal{F}}$ indicates averaging over all $\vec{\sigma}$.

3. The fraction of quasi-neutral directions $F_{\text{neut}}$ is a third measure of ruggedness. It was measured by computing the fraction of nearest neighbors on the landscape with function values that differed by less than a defined threshold ($10^{-4}$).

## Optimizability of linear landscapes

In this section, we demonstrate that a perfectly linear landscape is always optimizable by greedy search. A perfectly linear landscape is one that is perfectly fit by the linear model $\hat{\mathcal{F}}(\vec{\sigma}) = c + \sum_{i=1}^{S} a_i \sigma_i$ (and hence $r/s = 0$). We observe that a perfectly linear landscape can be optimized by simply adding each species with positive coefficient $a_i$ to the initial community and removing each species with negative coefficient $a_i$ from the initial community. A greedy search starting from any point on the landscape will do exactly this; over a sequence of search steps the search will add species with positive $a_i$ and remove species with negative $a_i$. This is because the contribution of a species to the community function on a linear landscape is independent of the current community context (position on the landscape).

Note that linearity is a sufficient, but not necessary, condition for greedy optimizability. For a deeper survey of discrete landscapes that are optimizable by greedy search, we refer the mathematically inclined reader to Refs. [101–103].

## Variance unexplained by higher-order models

We fit ecological landscapes of community function $\mathcal{F}(\vec{\sigma})$ incorporating higher-order species interactions using the following class of models:

$$\hat{\mathcal{F}}^{(k)}(\vec{\sigma}) = a^{(0)} + \sum_{i=1}^{S} a_i^{(1)} \sigma_i, + \sum_{i=1}^{S}\sum_{j=1}^{i-1} a_{ij}^{(2)} \sigma_i \sigma_j + \ldots \mathcal{O}(a^{(k)}) + \epsilon(\vec{\sigma}), \tag{14}$$

where the fit coefficients $a^{(0)}, a^{(1)}, \ldots$ defined up to order $k$ are obtained by least squares regression minimizing $(\hat{\mathcal{F}}^{(k)}(\vec{\sigma}) - \mathcal{F}(\vec{\sigma}))^2$. (Note that we can still use linear regression because the models are linear in the interaction coefficients.).

The fraction of variance unexplained by the model of order $k$, $U^{(k)}$, is given by

$$U^{(k)} = 1 - \frac{\sum_{\vec{\sigma}}(\hat{\mathcal{F}}^{(k)}(\vec{\sigma}) - \overline{\mathcal{F}})^2}{\sum_{\vec{\sigma}}(\mathcal{F}(\vec{\sigma}) - \overline{\mathcal{F}})^2} \tag{15}$$

where $\overline{\mathcal{F}}$ is the mean function; the numerator and denominator are the explained sum of squares and total sum of squares respectively. The number of higher order terms grows exponentially with the number of species, and so inverting matrices for linear regression became computationally expensive for large landscapes. Therefore, we used an alternative method to calculate the the fraction of variance unexplained for large landscapes based on Walsh decomposition (see S1 Text Sec. 3).

We imposed a threshold of $10^{-6}$ in the fraction of variance unexplained when plotting and fitting data in Fig 6 to be robust to numerical artifacts.

## Heuristic search protocols

We simulated community optimization attempts via a number of different search protocols.

**Heuristic gradient-ascent search and one-step reassembly.**    The gradient-ascent search proceeds as follows:

1. Choose a starting point on the landscape $A$, with initial composition $\vec{\sigma}^A$ and community function $\mathcal{F}(\vec{\sigma}^A)$.

2. Evaluate the relative change in steady-state community function at all points a single step away on the landscape, i.e, communities obtained by adding or removing a single species to the initial composition at $A$.

$$\Delta^B = \frac{\mathcal{F}(\vec{\sigma}^B) - \mathcal{F}(\vec{\sigma}^A)}{\mathcal{F}(\vec{\sigma}^A)}, \ \forall B : |\vec{\sigma}^B - \vec{\sigma}^A| = 1. \tag{16}$$

3. If the community function increased for any of the neighboring communities ($\Delta^B > 10^{-4}$), then we choose the neighboring point with highest community function and repeat from step 2 with this chosen community as $A$.

4. If the community function did not increase at any neighboring community, the search terminates. The current community is the community found by the search.

In Fig 10, we study a modified protocol, 'one-step reassembly' where search started from the best point out of $q$ randomly chosen communities. We also simulated a version of gradient-ascent search where search proceeded along the *first* direction with increasing community function (instead of the direction with maximal increase in community function). Similar results were obtained for this version and we omit discussion of these results for brevity.

**'bottleneck' protocol.**    The 'bottleneck' protocol proceeds as follows:

1. Grow $q$ random communities till steady state. Choose the community with highest function as the starting community.

2. $q$ replicates of the chosen community are generated.

3. $q - 1$ replicates are passaged via severe dilution shocks to generate stochastic species extinctions. One replicate is left unperturbed.

4. The $q$ communities are grown to steady state, and the community with the highest function is chosen.

5. Repeat steps 2–4 for a fixed number of iterations. The best community among the $q$ communities in the last iteration is the community found by the search protocol.

To simulate a dilution shock, the number of cells in the community $n_{cells}$ was first calculated by dividing by a fixed biomass conversion factor of $10^{-3}$. The number of cells in the community after dilution by factor $D$ was sampled from a Poisson distribution with mean $Dn_{cells}$. The number of cells of each species was then obtained by multinomial sampling with probabilities proportional to the biomass of each species.

**'addition + bottleneck' protocol.**    The 'addition + bottleneck' protocol closely follows the 'bottleneck' protocol above, with a single modification. In Step 3, after $q - 1$ replicates are subject to severe dilution shocks, a different randomly chosen species is added to each of these replicates. The $q^{th}$ replicate is still left unperturbed.

We simulated both dilution-based protocols ('bottleneck' and 'addition + bottleneck') for a range of dilution factors. The two strategies performed best at an intermediate dilution, where dilution stochastically eliminates a few species from the community, but not too many species

(S8 Fig). We report the results for the dilution factor that had the highest search efficacy in the main text (Fig 10).

To ensure a fair comparison between the dilution-based protocols and one-step reassembly, we fixed the number of replicates to $q = S + 1$ and the number of iterations to $S/2$ in the dilution-based protocols, and started one-step reassembly from the best of $q$ random communities in Fig 10. This ensured that the number of experiments in all strategies were similar.

Search efficacy of all search protocols was evaluated as the ratio between the community function at the end point of the search and the best community function on the landscape. We report search efficacy averaged over all possible starting points on the landscape in figures on the simple gradient-ascent search protocol. In Fig 10, which compared the dilution-based protocols and 'one-step reassembly' we report the search efficacy averaged over 10 independent searches on each landscape.

## Sampling ecological landscapes

We estimated ruggedness measures from appropriately sampled subsets of the community function landscapes. To estimate $r/s$, we randomly sampled a fraction, $f$, of the possible $2^S$ communities. The linear model was fit on the sampled data points to estimate $r/s$. Since the model has $S + 1$ parameters, we need to sample at least $S + 1$ points. To estimate $1 - Z_{nn}$ and $F_{\text{neut}}$, we sampled a fraction $f$ of all communities by choosing half of the communities randomly and generating one random nearest neighbor for each of them. $1 - Z_{nn}$ and $F_{\text{neut}}$ were then calculated from the sampled points.

## Analysis of experimental data

The experiments by Langenheder, et. al. measured the time series of cumulative metabolic activity, through the intensity of a dye, for each community and environment [32]. We obtained the rate of metabolic activity from the data via a linear fit of the metabolic activity over the last six time points. The communities had reached a steady state in metabolic activity, as evidenced by the high $R^2$ of the linear fits (S10 Fig). We analyzed the landscape formed by the steady-state metabolic rate in Fig 11. We imposed a threshold in the fraction of variance unexplained ($10^{-4}$) when plotting and fitting data in Fig 11B.

The experiments in Clark, et al. [65] measured OD and production of four organic acids in different combinations of 25 different species. To generate the landscape, we used measurements from non-contaminated experimental points (as classified by the paper), which provided us data from 577 unique species combinations. Measurements corresponding to experimental replicates from the same species combinations were averaged to generate the landscape. $r/s$ was estimated using the same procedure as in simulated landscapes. The procedure for parameterizing the butyrate production function is described in the Methods section describing community functions.

## Supporting information

**S1 Text. Supplementary text containing further mathematical and simulation details, other ruggedness measures, and additional information regarding to literature cited in Table 1.**
(PDF)

**S1 Fig. Niche overlap between species increases with number of resources consumed.** The niche overlap in the 16 species pools shown in Fig 2. Niche overlap was quantified by the average cosine similarity between the consumption vectors of each species pair in the pool, as in previous studies [33]. In these simulations, each non-zero consumption matrix element was drawn from a gamma distribution. The niche overlap increased with number of resources when consumption matrix elements were drawn from other distributions as well. See S11 Fig.
(PDF)

**S2 Fig. $\vec{N}(\vec{\sigma})$ is linear for specialists and nonlinear for generalists—independent of the number of resources and presence of cross-feeding.** The map from species presence to steady-state abundances $\vec{N}(\vec{\sigma})$ becomes more nonlinear with increasing niche overlap across a range of models and parameters. The linearity of the map is quantified by $\overline{R^2}$. Niche overlap increases when species in the pool consume more resources. We plot consumer resource models with $S = 16$, $M_{tot} = 16$ in **(A)**, $S = 16$, $M_{tot} = 32$ in **(B)**, and $S = 16$, $M_{tot} = 8$ in **(C)**; and cross-feeding model with $S = 12$, $M_{tot} = 12$ in **(D)**. There were 10 replicates for each data point. Simulation parameters are described in Methods.
(PDF)

**S3 Fig. The ruggedness of ecological landscapes of other community functions.** Panels show $r/s$, $1 - Z_{nn}$, and $F_{neut}$ the landscapes with community function being pair productivity, butyrate production, and focal species abundance. The fraction of neutral directions $F_{neut}$ was high at low niche overlap, when all of the species occupied separate niches, because the species (or species pair) responsible for the community function was unaffected by the addition or removal of other species. This causes the community function to be left unchanged and a concordant increase in the number of neutral directions. Simulations were the same as in Fig 5.
(PDF)

**S4 Fig. Estimating search efficacy from noisy data using landscape ruggedness.** Magnitude of the correlation between ruggedness estimated from noisy data and search efficacy remains high even when experimental noise is as large as the community function itself. Two forms of noise were simulated, additive noise in measurement **(A)** and multiplicative noise **(B)**. The strength of the noise $\lambda_M$ and $\lambda_X$ is measured relative to the community function; therefore a noise strength of one means that the contribution from noise is as large as the community function itself. The community function was the productivity of a pair of species in panel A and diversity in panel B. Noise was simulated as described in SI. Data was obtained from simulations in Fig 5.
(PDF)

**S5 Fig. Niche overlap increases ruggedness and difficulty of search in the presence of cross-feeding. (A,B)** Search outcome and ruggedness of a cross-feeding model where the niche overlap between species was varied by changing the number of resources consumed by each species, mirroring results obtained in model without cross-feeding (Figs 3 and 5). **(C)** Ruggedness remained informative of search efficacy. Community function was the abundance of a focal species. Parameters are described in Methods.
(PDF)

**S6 Fig. Ruggedness and search efficacy in the crossfeeding model.** Panels demonstrate the search efficacy, ruggedness, and correlation between search efficacy and ruggedness on landscapes with community functions of butyrate production, diversity, and pair productivity. All

reported correlations were statistically significant.
(PDF)

**S7 Fig. Landscape ruggedness is informative of search efficacy at different levels of environmental complexity. (A,B)** Search efficacy and ruggedness of a cross-feeding model where the number of resources supplied to the community was varied. **(C)** Ruggedness remained informative of search efficacy. Community function was the abundance of a focal species. The total amount of resources supplied was held fixed in these simulations with parameters as described in Methods. Simulations where the amount of each resource supplied was held fixed instead gave similar results.
(PDF)

**S8 Fig. Performance of dilution-based search protocols is optimal only for a small range of dilution factors.** The dilution-based search protocols, 'bottleneck' and 'addition + bottleneck', have a high search efficacy only if the bottlenecking step kills a few species but not too many. Therefore, it works best for a narrow range of dilution factors where only order 10 cells survive bottlenecking, before being subject to invasion [79]. Community function was the productivity of a pair of species and simulations were the same as in Fig 9.
(PDF)

**S9 Fig. $\overline{R^2}$, is unaffected by rounding negative predictions of the linear model to zero.** $\overline{R^2}$ after rounding negative abundance predictions to zero is in good agreement with $\overline{R^2}$ computed without rounding negative predictions of the linear model shown in Fig 2.
(PDF)

**S10 Fig. Rate of metabolic activity is constant over the duration of the experiment.** While the experiment by Langenheder et. al. [32] assayed the cumulative metabolic activity of the microbial communities at different timepoints, used the rate of metabolic activity as the community function. The rate of metabolic activity measured as the slope of a linear fit to the cumulative metabolic activity. The rate was at steady state as evidenced by the high $R^2$ of linear fit to the data as shown in the figure.
(PDF)

**S11 Fig. Niche overlap between species increases with number of resources consumed for different distributions of consumption matrix elements.** The niche overlap in 16 species pools where the non-zero consumption matrix elements were sampled from a uniform distributions **(A,B)**, and lognormal distributions **(C,D)**. Niche overlap was quantified by the average cosine similarity between the consumption vectors of each species pair in the pool, as in previous studies [33]. The uniform distribution in A extended from 0.5 to 1.5; in B extended from 0.1 to 20.5. The lognormal distribution parameters were $\mu = 0$, $\sigma = 0.3$ in panel C and $\mu = 0.6$, $\sigma = 0.6$ in panel D. The extent of increase in niche overlap reduces as the variability in the sampling decreases.
(PDF)

**S12 Fig. Ruggedness remains informative of search efficacy even if initial community was fixed.** For Shannon diversity, the ruggedness measure is informative not only about the average search outcome **(A)**, but also about search outcome starting from the community with all species present **(B)**, a randomly chosen species in monoculture **(C)**, and from a randomly chosen community with half the candidate species present **(D)**. Simulation data was the same as in Fig 3.
(PDF)

## Acknowledgments

The authors would like to thank Pankaj Mehta, Gabriel Birzu, Shreya Arya, and James O'Dwyer for helpful discussions and comments, Silke Langenheder for sharing experimental data. and Robert Marsland for sharing illustrations of the consumer-resource model. Simulations were carried out on the Boston University Shared Computing Cluster.

## Author Contributions

**Conceptualization:** Ashish B. George, Kirill S. Korolev.

**Data curation:** Ashish B. George.

**Formal analysis:** Ashish B. George.

**Funding acquisition:** Kirill S. Korolev.

**Investigation:** Ashish B. George.

**Methodology:** Ashish B. George.

**Supervision:** Kirill S. Korolev.

**Visualization:** Ashish B. George.

**Writing – original draft:** Ashish B. George, Kirill S. Korolev.

**Writing – review & editing:** Ashish B. George, Kirill S. Korolev.

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
