## [Decision Letter · Decision Letter 0]

23 May 2022

Dear Dr. Bino George,

Thank you very much for submitting your manuscript "Ecological landscapes guide the assembly of optimal microbial communities" for consideration at PLOS Computational Biology.

As with all papers reviewed by the journal, your manuscript was reviewed by members of the editorial board and three independent reviewers. In light of the reviews (below this email), we would like to invite the resubmission of a significantly-revised version that takes into account the reviewers' comments.

In particular, Reviewer 1, while largely positive about the study, raises questions about the effect of niche creation through by-product secretion during community assembly on the results, the measure of resource overlap, and validity of the experimental results (indeed, perfect ant-correlation with 3 points does not permit any reliable conclusions). Reviewer 2 questions the generality of the results, and asks for greater theoretical support for several choices made, including choice the the community function measure. They too question the value of the experimental evidence provided, and the relevance of the theory to it. Reviewer 3, in a similar vein as 2, asks for some robustness analyses, and a more rigorous treatment of the optimization problem underlying maximisation of microbial community functioning. And like Reviewer 1, they too raise the issue of niche creation through by-product secretion during community assembly. 

We cannot make any decision about publication until we have seen the revised manuscript and your point-by-point responses to the reviewers' comments. Your revised manuscript is also likely to be sent to reviewers for further evaluation.

Sincerely,

Samraat Pawar

Guest Editor

PLOS Computational Biology

Kiran Patil

Deputy Editor

PLOS Computational Biology

Reviewer's Responses to Questions

Comments to the Authors:

Please note here if the review is uploaded as an attachment.

Reviewer #1: This is an interesting paper that I very much enjoyed reading. I believe it represents an important contribution to the field of microbial ecology. As the authors mention, a number of protocols for the search of optimal consortia have been proposed in the past, but have only been very moderately successful at best. Cementing the ecological landscape framework could be key to develop better strategies, and in that sense this work provides useful tools to understand when and how ecological landscapes can be navigated.

I do have some concerns, particularly regarding what the authors refer to as the "density of interactions", the metric they use to quantify it, and the claim of it shaping ecological landscapes. Specific comments follow:

1. In figure 2 and in the entire first section of the Results ("Niche overlap controls the complexity of ecological landscapes") it was not clear to me which model the authors were using (with/without cross-feeding) until the very last sentence of the section. I think it would be helpful to specify this earlier on and in the figure legend. Similarly, I think it would help to describe both models at least minimally in the Model section and not only in the Methods. Specifically, I had to look in the Methods to find that, in the model with cross-feeding, a single resource was supplied externally (versus all resources supplied externally in the model with no cross-feeding). This is completely reasonable, but since I think the difference affects how one should interpret the results and the metrics that are used to describe the ecological properties of the system, I would say it is important to at least mention it in the main text.

2. The concept of more "overlap in resource utilization profiles" leading to a higher "density of interactions" (lines 144-146) makes intuitive sense to me. However, I think there might be some nuance to it. If I am understanding it correctly, the authors refer to the idea that, if a resource can be consumed by multiple species, there will be competitive interactions among them for said resource and therefore a higher interaction density is expected in the 'generalist' case versus the 'specialist' case (if this is correct, I think explicitly explaining it in the main text would help). However, the role of facilitative interactions (in the model with cross-feeding) in this framework is unclear to me. I could see how, in the model with cross-feeding, a higher degree of specialization could lead to species secreting byproducts that they cannot utilize themselves, thus creating new niches into which additional consumers could be recruited (see e.g. Estrela et al. Diversity begets diversity under microbial niche construction, biorXiv, 2022). Then, one could make the argument that increasing the degree of consumer specialization could increase the density of *facilitative* interactions (and not only decrease the density of *competitive* interactions). It is not obvious to me why one would not expect a high interaction density in the model with cross-feeding with highly specialized consumers, even if these interactions were predominantly facilitative. From this perspective, one could maybe intepret the results in the first section of this paper (and in figure S1) as indicating that "community assembly becomes more predictable as interactions become less competitive" rather than "as the number of interspecific interactions decreases". In summary, I think that the effect of changing the degree of consumer specialization level could have different effects on the two models in terms of a generic "density of interactions" if one accounts for both competition and facilitation. I might be understanding it wrong, but this was not straightforward for me to understand and I would appreciate if the authors could elaborate on it.

3. I am not sure the ratio between M_consumed and M_total is generally a good proxy for the overlap in resource utilization. If I am correct, the more species that *can* consume a resource, the higher the number of *potential* interactions. But this can depend on, first, how species "choose" to consume resources (if at all), and, second, how much variation in the values of the uptake rates there is (i.e. the width of the distribution from which they are drawn).

In the absence of species-specific resource preference profiles, species essentially consume all resources that they can uptake. But if resource preferences are variable across consumers, species can preferentially utilize a specific resource while it is available, even if they are also able to consume a different one (and, in the more general case, they can exhibit higher preference for a resource that is not necessarily that for which they have a higher uptake rate). Even assuming no species-specific resource preference profiles, if the variation in the uptake rates is large, it could still be the case that any two species could have similar profiles in terms of the resources they *can* consume (i.e. which uptake rates are non-zero) while still exhibiting substantial differences in the actual values of their uptake rates. In either of these scenarios I could imagine that, once equilibrium is reached, the overlap in resource consumption in the community could be lower than the overlap in which uptake rates are non-zero. In other words, it could happen that species are very similar in terms of the resources they can consume (M_consumed close to M_total) but in practice utilize different resources in the stabilized communities due to differences in their preference profiles or the values of their uptake rates being very unevenly distributed.

I understand that the simulations in this work are carried out under the assumption of no regulation or species-specific resource preference profiles. The implementation of resource preference in consumer-resource models is a problem on its own (e.g. Wang, Goyal et al., Nat Commun 2021) that falls out of the scope of this paper, and I certainly do not think it is necessary for the authors to tackle this issue in their work. I am, however, left wondering if M_consumed/M_total is a good proxy for the overlap in resource utilization profiles in any general scenario. It could be useful to analyze the effect of changing the width and/or skewness of the distribution used to sample uptake rates, or to consider alternative metrics for resource utilization overlap. Alternatively, the authors could further elaborate in the text on why they think M_consumed/M_total is an appropriate metric of niche overlap under their conditions.

4. In lines 334-335, the authors write "the search efficacy was perfectly anti-correlated with ruggedness on the three pure carbon sources (Spearman's correlation coefficient rho = -1, p = 0.0)". I'm not sure that a perfect Spearman's correlation of only 3 data points and such small variations in both axis (as the authors themselves point out) is really noteworthy. In fact, even if no correlation was observed, the author's argument would not be invalidated --it could still be the case that other ecological landscapes (perhaps with a different choice of species, environments, and/or functions, as the authors mention) could be more rugged and difficult to navigate. The question that comes to mind looking at figure 10 is how rugged empirical ecological landscapes are. I understand that the authors only looked at the specific dataset from Langenheder et al. because it is a combinatorially complete landscape, but in figure 7 they show that they can obtain good estimates of ruggedness from partial data. Even if they cannot compute search efficacy in a partial landscape, I think it would still be useful to have a sense of how rugged ecological landscapes are in a wider context. Some examples of partial landscapes that I suggest the authors check are:

Clark et al. Design of synthetic human gut microbiome assembly and butyrate production. Nat Commun 12:3254 (2021)

Kehe et al. Massively parallel screening of synthetic microbial communities. PNAS 116(26):12804-12809 (2019)

Reviewer #2: In the submitted manuscript, George and Korolev develop a framework to analyze heuristic strategies used to determine the composition of microbial communities that optimize a given ecological function.

While the content of the manuscript is mostly scientifically valid (with some caveats, see points below), the results are not of high importance to the researchers in the field. In fact, the results found by the authors rely on several arbitrary choices (e.g., which ecological function to optimize) and it is not obvious how their conclusions would generalize to different choices. Therefore, publication in PLoS Computational Biology cannot be recommended.

The manuscript would, however, be suitable for publication in other journals like PLoS ONE, provided that the points covered below are appropriately addressed:

1) Some of the references cited by the authors in Table 1 (e.g., [27] and [28]) do not support their claims. The authors should choose more carefully which references to use in that Table.

2) In lines 184-185, the authors state “The precise definition of the diversity did not affect our conclusions, so we focused on the Shannon Diversity”. This statement is not substantiated by any result. The authors should provide concrete examples that their statement is indeed true (i.e., show that their results do not change with different definitions of diversity).

3) In several parts of the manuscript (e.g., Fig. 3, Fig. 6) the authors show results relative to the ecological functions “pair productivity” and “focal abundance”. Contrarily to “diversity” (the other one they consider), these two functions depend by definition on the choice of one or two particular species in the community. The authors, however, never specify if the results they are showing are relative to a particular choice of species or if they are an average over different choices. This fact should be stated clearly whenever they show results relative to “pair productivity” and “focal abundance”.

4) In Fig. 4 and lines 229-231, the authors claim that all metrics of ruggedeness increase with niche overlap. However, this is clearly not true for Fig. 4B since the metric quickly saturates to a constant value. This discrepancy should be addressed more carefully in the manuscript. Furthermore, the authors show in Fig. S2 the equivalent of Fig. 4 for another ecological function (pair productivity) but never show the same results for focal abundance. This result should be included in the manuscript to substantiate the author’s claims, and all the plots should be put together in the Main Text instead of being separated between Main Text and Supplementary Information.

5) The plot shown in Fig. 5 is not really a surprising result, since it’s showing that increasing the order of the function used to fit a model increases the accuracy of the results. This plot is a nice control of the validity of the author’s results, and should be moved to the Supplementary Information instead.

6) In the paragraph starting at line 271, the authors claim that their results generalize to other models and search protocols. While what they show substantiate this claim for different search protocols, the authors are not really considering different models: they are simply keeping the same model they’ve been using in the rest of the manuscript (i.e., the consumer-resource model) and modified with the addition of cross-feeding. However, this is still a consumer-resource model and not a different one. The authors should change the wording of that paragraph and state that the results generalize to consumer-resource models with cross-feeding (and not “to other models” in general).

7) In Fig. 8B, the search efficacy exhibit a very large variation for a fixed value of the metabolic leakage l (e.g., the distribution shown for l=0.2 goes from 25% to 100%). This fact should be discussed in detail by the authors, since such a large variation could make the heuristic search strategy worthless. Furthermore, the results shown in this figure are relative to only one of the ecological functions that the authors consider in the manuscript (i.e., the focal abundance). The authors do not show anywhere in the manuscript (including the Supplementary Information) the results for the other two ecological functions they consider. In order to substantiate their claims, these results should be shown alongside Fig. 8.

8) The application of the model to experimental data that the authors show in the paragraph starting at line 315 is very weak: by the author’s own admission, the results are not statistically significant. This part of the manuscript should either be removed or supported with a more carefully executed experiment. For example, the dataset they use contains only six species, while all the results that the author showed about the efficacy of the heuristic search strategies are relative to communities with 16 species. Since the dimensionality of the system’s landscape changes dramatically if we have 6 or 16 species, the authors should discuss in detail how a different species pool size affect the application of their model to experimental data.

Furthermore, in lines 331-333 the authors state “Because of the rather simple metabolic interactions in these communities, the community landscapes were quite smooth and the search efficacy was quite high”. This claim, however, is not substantiated by any result.

9) In the paragraph starting at line 402, the authors show how they have chosen the parameters used in the simulations. Several of those parameters have been drawn from particular distributions (e.g., gamma distributions). While this choice per se is not problematic, the authors should explain why they have made this choice, and show what happens if the parameters are drawn from other distributions (e.g., a normal, log-normal or uniform distribution).

10) In line 450, the authors state “This truncation did not affect our results” but do not show any result that substantiates this claim.

Reviewer #3: George and Korolev studies the effectiveness of heuristic gradient-ascent type approaches to search species in microbial communities in order to optimize community functions. Using simulations from consumer resource models, they demonstrate that (1) search effectiveness is inversely related to the ruggedness of the underlying ecological landscape, (2) communities with increased niche overlap tend to be more rugged, and (3) ruggedness can be estimated with limited and noisy data. I think the current manuscript is difficult to follow without prior exposure to the concept of landscape ruggedness and I suggested several ways to improve clarity and to make connection with established concepts in optimization theory (Major comments 1, 5, 6, 8). There are also additional robustness analysis that I hope the authors could perform to improve relevance to practice (Major comments 2, 3).

Major comments:

1. The clarity of this manuscript could be enhanced by including a step-by-step flow chart of how ruggedness is computed. The current description of ruggedness is either too vague or too detailed.

2. I would encourage the authors to perform some additional robustness analysis of the results in Figure 7. This includes, for example, considering models that include non-resource mediated interactions, such as pH-mediated interactions. These interactions are described by non-monotone functions hence may challenge the ability to predict ruggedness (or roughly speaking convexity of optimization problem) from limited data.

3. Related to my previous comment, the consumer resource model used in this paper does not account for sequential resource uptake or growth regulation by multiple metabolites/resources. These growth models are common for bacterial growth and considering them could demonstrate the robustness of the results and significantly improve the relevance of these computational results to experiments.

4. The summary in Table 1 lacks mathematical rigor and needs to be seriously re-considered. For example, multi-dimensional Lotka-Volterra models always have multiple steady states, as opposed to “single steady state” shown in the table. While there is only one positive equilibrium, the other equilibria with at least one species being 0 (i.e., extinct) could still be stable. “Typically a single steady state” needs to be clarified. Does it mean the number of steady states depends on parameter? Additionally, what is relevant to practice is the number of stable steady state instead of the number of steady states. This needs to be clarified in the table. In general, I think the contents in Table 1 are over-generalizing the complicated behaviors that could arise from nonlinear dynamical systems and needs to be made precise mathematically if the authors want to keep them in the manuscript.

5. While the inverse relationship between ruggedness and search efficacy is clear through numerical simulations, the quality of this paper could be significantly improved if the authors could provide some analytical insights to this phenomenon, especially by making connections to concepts in optimization.

If I understand correctly, in its very basic form, most of the ruggedness measures of F quantify how close the function F could be approximated by a linear function of the absence/presence of species (i.e., vector \\sigma). One of the main results in this paper is that the objective function evaluated at the steady state of the consumer resource model with small ruggedness (hence close to a linear function) tend to be easier to optimize using a heuristic search that resembles gradient-ascent. This result could be investigated/discussed in the context of optimization theory. In particular, consider the ideal case where the objective function is linear, is there any theory/heuristic from integer programing literature that suggests convergence of the gradient-ascent algorithm to global maximum? More generally, how ruggedness is connected to the convexity of an optimization problem, which I think is a more well-defined/well-known mathematical property compared to ruggedness? Perhaps such connections have been made clear already in the literature and the authors are encouraged to discuss/describe them in this manuscript. This will significantly help understanding and diminish ambiguity.

6. The inverse relationship between resource overlap and the ability of the linear model (1) to predict community assembly outcome is interesting. Is it possibly to develop some physical intuition of this using analytical solutions to the consumer resource model for a simple two-species community with and without niche overlap?

7. The authors may want to discuss alternative search methods that could be more effective in a rugged landscape.

8. The clarity of this manuscript could be benefited substantially by moving some key mathematical contents, such as the consumer resource model and the definition of ruggedness in the Methods section to the main text. The descriptions of ruggedness by words in the main text is very difficult to follow. In addition, I strongly encourage the authors to include a precise mathematical formulation of the optimization problem at hand (e.g., F is the objective function, \\sigma vector, taking values in {0,1}^S is to be optimized, and subject to the consumer resource dynamics). The current description of this core optimization problem in line 112-125 makes it difficult to link physical properties to mathematical notations.

Minor comments:

1. Is niche overlap discussed in line 149-153 affected by the magnitude of parameters in the consumption matrix?

2. The model in equation (1) should not contain the “model error” part (\\epsilon_i). This is only needed if the model itself is stochastic. However, here the model is deterministic and fitting residue should not be considered part of the model.

3. The connection between N in line 136 and the general function F is unclear. My understanding is that N is an instantiation of F. If this is correct, please state it explicitly.

4. Line 157-158: Please indicate what data are used for fitting (i.e., steady state abundance generated from simulation consumer resource models, if I understand correctly).

5. Line 166: I do not understand what “community assembly becomes simpler” means. Does it mean that steady state community composition becomes easier to predict using a linear model of species absence/presence?

6. What does the color bar in Figure 2A-B (abundance predicted vs. observed panels) represent?

7. In line 109-111, the statement that “multistability are more of an exception” in microbial communities is rather controversial and may need to be toned down. The authors are also encouraged to discuss how the current method fails in the presence of multi-stability.

8. What does the color shades in Figure 6D mean?

9. Fig. 10A and Fig. 10B could switch position since 10B is referred first in the text.

10. The definition of \\bar{F} is unclear in equations (11) and (13). What is the mean function taking average over? Is it averaging the objective function F evaluated at all possible values of \\sigma?

Have the authors made all data and (if applicable) computational code underlying the findings in their manuscript fully available?

Reviewer #1: No: Extracted from the review file: "The authors will share the code used to generate and analyze the data in the manuscript upon manuscript publication"

Reviewer #2: No: The code was not available with the manuscript

Reviewer #3: No: I did not find sufficient information about sharing of code and data provided in the manuscript

PLOS authors have the option to publish the peer review history of their article (what does this mean?). If published, this will include your full peer review and any attached files.

Do you want your identity to be public for this peer review? For information about this choice, including consent withdrawal, please see our Privacy Policy.

Reviewer #1: Yes: Juan Diaz-Colunga

Reviewer #2: No

Reviewer #3: No

Figure Files:

Data Requirements:

Reproducibility:

To enhance the reproducibility of your results, we recommend that you deposit your laboratory protocols in protocols.io, where a protocol can be assigned its own identifier (DOI) such that it can be cited independently in the future. Additionally, PLOS ONE offers an option to publish peer-reviewed clinical study protocols. Read more information on sharing protocols at https://plos.org/protocols?utm_medium=editorial-email&utm_source=authorletters&utm_campaign=protocol

---

## [Editor Report · Decision Letter 1]

27 Jul 2022

Dear Mr Bino George,

Thank you for submitting your revision.

However, the submission is incomplete because you have not shared the code used to generate and analyze the data in the manuscript. This is in line with the PLOS guidelines and was also raised by all three reviewers.

Please do so within 60 days. If you anticipate any delay, please let us know the expected resubmission date by replying to this email. Please note that revised manuscripts received after the 60-day due date may require evaluation and peer review similar to newly submitted manuscripts.

Sincerely,

Samraat Pawar

Guest Editor

PLOS Computational Biology

Kiran Patil

Deputy Editor

PLOS Computational Biology

---

## [Decision Letter · Decision Letter 2]

13 Sep 2022

Dear Dr. Bino George,

We are pleased to inform you that your manuscript 'Ecological landscapes guide the assembly of optimal microbial communities' has been provisionally accepted for publication in PLOS Computational Biology.

Best regards,

Samraat Pawar

Guest Editor

PLOS Computational Biology

Kiran Patil

Section Editor

PLOS Computational Biology

We have reviewed the revised manuscript. The authors have satisfactorily addressed all three Reviewers' comments and suggestions, significantly expanding their results in the process, including the addition of a new dataset on a different community function.

Reviewer's Responses to Questions

**Comments to the Authors:**

Reviewer #1: The authors have addressed all of my comments, and seem to have addressed those of the other reviewers as well. They have substantially expanded their results, with the addition of a new data set corresponding to a different community function. They also added some new figures to the main text and the supplementary material, which address my main concerns and make the manuscript easier to follow. Overall, very nice work.

**Have the authors made all data and (if applicable) computational code underlying the findings in their manuscript fully available?**

Reviewer #1: Yes

PLOS authors have the option to publish the peer review history of their article (what does this mean?). If published, this will include your full peer review and any attached files.

Reviewer #1: **Yes: **Juan Diaz-Colunga

---

## [Editor Report · Acceptance letter]

28 Sep 2022

PCOMPBIOL-D-22-00529R2 

Ecological landscapes guide the assembly of optimal microbial communities

Dear Dr B. George,

I am pleased to inform you that your manuscript has been formally accepted for publication in PLOS Computational Biology. Your manuscript is now with our production department and you will be notified of the publication date in due course.

With kind regards,

Marianna Bach
